# Synergistically optimized electron and phonon transport in high-performance copper sulfides thermoelectric materials via one-pot modulation

Yi-Xin Zhang[1], Qin-Yuan Huang[1], Xi Yan[1], Chong-Yu Wang[1], Tian-Yu Yang[1], Zi-Yuan Wang[1], Yong-Cai Shi[1], Quan Shan [1], Jing Feng[1] & Zhen-Hua Ge [1] ✉

Optimizing thermoelectric conversion efficiency requires the compromise of electrical and thermal properties of materials, which are hard to simultaneously improve due to the strong coupling of carrier and phonon transport. Herein, a one-pot approach realizing simultaneous second phase and Cu vacancies modulation is proposed, which is effective in synergistically optimizing thermoelectric performance in copper sulfides. Multiple lattice defects, including nanoprecipitates, dislocations, and nanopores are produced by adding a refined ratio of Sn and Se. Phonon transport is significantly suppressed by multiple mechanisms. An ultralow lattice thermal conductivity is therefore obtained. Furthermore, extra Se is added in the copper sulfide for optimizing electrical transport properties by inducing generating Cu vacancies. Ultimately, an excellent figure of merit of ~1.6 at 873 K is realized in the $Cu_{1.992}SSe_{0.016}(Cu_2SnSe_4)_{0.004}$ bulk sample. The simple strategy of inducing compositional and structural modulation for improving thermoelectric parameters promotes low-cost high-performance copper sulfides as alternatives in thermoelectric applications.

Thermoelectric (TE) conversion technology is capable of realizing power generation and small refrigeration[1–3]. The excellent conversion efficiency of the thermoelectric materials that are the key component of TE devices is the main parameter by which to evaluate the potential candidates[4–6]. Other requirements, including service stability, mechanical performance, cost, and toxicity, are essential to promote thermoelectric materials in device assembly and extensive application[7]. The conversion efficiency is gauged by the thermoelectric dimensionless figure of merit: $ZT = S^2\sigma T/\kappa$, where $S$, $\sigma$, $T$ and $\kappa$ are the Seebeck coefficient, electrical conductivity, absolute temperature and total thermal conductivity, respectively[8]. It is difficult to simultaneously optimize the electrical and thermal transport properties because of the complex coupling of various TE parameters[9,10]. Introducing a second phase by in situ precipitates[11,12,13] or phase separation[14,15] has been proven to be effective in restraining the lattice thermal conductivity, which can be individually improved. Decoupling other TE parameters usually involves complex and multistep optimization strategies[16,17]. In addition, the service stability of thermoelectric materials must be considered, which requires the maintenance of the composition and microstructure of TE materials under external temperature or an electrical field.

Copper sulfides have been studied in the TE field for more than 190 years, which possess ultralow lattice thermal conductivity because of the strong phonon scattering effects and the vanished transverse wave phonon vibration caused by disordered Cu ion migration[18]. Lots of efforts have been put to improve the electrical transport properties while maintaining the low thermal conductivity of copper sulfides. The typical strategies are composition off-stoichiometry[19,20], element-

[1]Faculty of Materials Science and Engineering, Kunming University of Science and Technology, Kunming 650093, China. ✉e-mail: zge@kust.edu.cn

doping[21–25] and second phase compositing[26–28]. The hole concentrations of Cu$_{2-x}$S can vary in at least two orders of magnitude on tailoring the contents of Cu vacancies[19,29]. By contrast, foreign atoms are hard to enter into the lattice of superionic conductor materials, thus the effect of extrinsic doping on carrier concentration adjustment is not obvious[30]. In addition, compositing with second phase benefits to further enhance the phonon scattering for reducing thermal conductivity, extra particles are also capable of blocking long-range migration of Cu ions for improving stability of copper sulfides[31]. It has been reported that both Sn doping and Se alloying could improve the thermoelectric performance of copper sulfides owing to the porous microstructure and tuned bonding energy[32–35]. Previous works have also investigated that compositing with multi-walled carbon nanotube[28] or graphene[26] are effective in improving thermoelectric performance of copper sulfides by decreasing lattice thermal conductivity. Additionally, Cu$_{2-x}$S phase-junction nanocomposites with superior thermoelectric performance can be synthesized by surface-ligand tuning[36], indicating that carefully adjusting Cu vacancies benefits to optimize the thermoelectric properties. Nevertheless, the electrical stability of these materials has not been studied in detailed, and the method of spontaneously introducing Cu vacancies and in-situ generated precipitates according to the designed composition by one-pot approach have not reported yet.

Herein, a one-pot modulation strategy for simultaneously adjusting carrier and phonon transport in copper sulfides is proposed as shown in Fig. 1. The first step aims at in-situ generating Cu$_2$SnSe$_4$ precipitates and various lattice defects by adding a suitable content of Sn and Se in Cu$_2$S, benefiting for strengthening the phonon scattering. The lattice thermal ($\kappa_l$) conductivity at 873 K is therefore significantly lowered (Fig. 1b). Second, additional Se is added to further introduce the Cu vacancies and reduce the bond energy in the material. Tuning the Cu content can regulate the hole concentration and the temperature of the phase transition, then the highly enhanced average power factor ($PF_{ave.}$) is therefore realized (Fig. 1c). Cu vacancies are the key factors of optimizing the electrical conductivity and stability of copper sulfides, whereas the multiscale lattice defects act as the key roles in overall reducing lattice thermal conductivity of the materials. Compositional regulation and structural evolution are simultaneously realized by the two-step optimization, benefiting for synergistically improving thermoelectric performance. As a result, the $ZT$ of the Cu$_{1.992}$SSe$_{0.016}$(Cu$_2$SnSe$_4$)$_{0.004}$ specimen reaches approximately 1.6 at

873 K, which is comparable to that of other Cu-based thermoelectric materials (Cu$_2$Se, Cu$_2$Te) at the same temperature (Fig. 1d)[19,24,37–40]. Furthermore, the electrical stability and mechanical performance of this sample are significantly enhanced. This strategy may also work for enhancing thermoelectric performance in other superionic conductors.

## Results and discussion

The phase structures of the Cu$_2$S, Cu$_{1.992}$S(Cu$_2$SnSe$_4$)$_{0.004}$ and Cu$_{1.992}$SSe$_{0.016}$(Cu$_2$SnSe$_4$)$_{0.004}$ bulk specimens were characterized by X-ray diffraction (XRD), as shown in Fig. 2. The diffraction peaks of the pristine sample can be indexed to the monoclinic Cu$_2$S phase (PDF#83-1462). Furthermore, the main phase changes to tetragonal Cu$_{1.96}$S (PDF#29-0578) after introducing a suitable content of Sn and Se by one-pot modulation, indicating the generation of Cu vacancies in the matrix. The added Sn and Se prefer to consume Cu to form the Cu$_2$SnSe$_4$ second phase and generate a tiny content of Cu vacancies. There are diffraction peaks of Cu$_2$SnSe$_4$, verifying that the reaction occurred during the synthesis process. Notably, the characteristic peaks of the Cu$_{1.992}$SSe$_{0.016}$(Cu$_2$SnSe$_4$)$_{0.004}$ specimen shift toward a higher $2\theta$ than those of the Cu$_{1.992}$S(Cu$_2$SnSe$_4$)$_{0.004}$ sample. The lattice shrinkage implies that more Cu vacancies are produced after adding additional Se. Excess Se can easily enter into the lattice of copper sulfides matrix, which have been proved in the previous studies[32,39]. The relative density of the bulk materials synthesized by one-pot modulation first decreases and then gradually increases with additive content, indicating the generation of a tiny content of pores, which might then be filled in by precipitates. The density of the bulk materials remains at a high level (Supplementary Fig. 1 and Supplementary Table 1).

The fracture morphology of all samples is exhibited in Supplementary Fig. 2. The transgranular fracture of the pure sample gradually shifts to intergranular fracture with increasing addition content, corresponding to the stratiform fractured surface changing to particle-like grains. Additionally, pores are generated along the grain boundaries and within the grains for the samples fabricated by one-pot modulation, which is mainly ascribed to the partial sulfur volatilization in the Cu-deficient copper sulfides. The content of pores in the materials is in good agreement with the change in density. Subsequently, microscale precipitates can be visualized in the material, which mainly exist along the grain boundaries. An electron probe microanalyzer

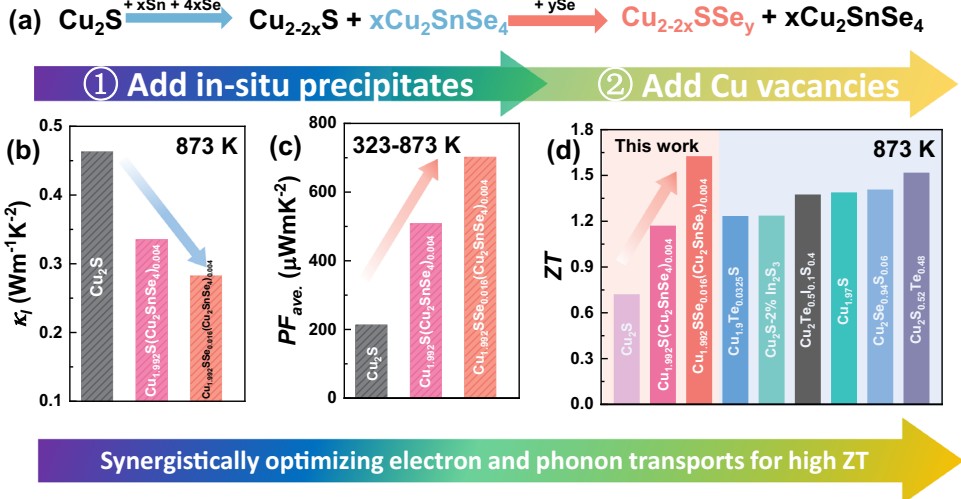

**Fig. 1 | The designed one-pot strategy of utilizing second phase and Cu vacancies modulation to optimize thermoelectric properties of copper sulfides. a** The process of a one-pot modulation strategy in adjusting electrical and thermal transport properties. **b** Lattice thermal, (**c**) average power factor and (**d**) $ZT$ of Cu$_2$S, Cu$_{1.992}$S(Cu$_2$SnSe$_4$)$_{0.004}$ and Cu$_{1.992}$SSe$_{0.016}$(Cu$_2$SnSe$_4$)$_{0.004}$ bulk samples. Peak $ZT$ values at 873 K for other copper-based thermoelectric materials are added for comparison, and the data were taken from refs. 19,24,37–40.

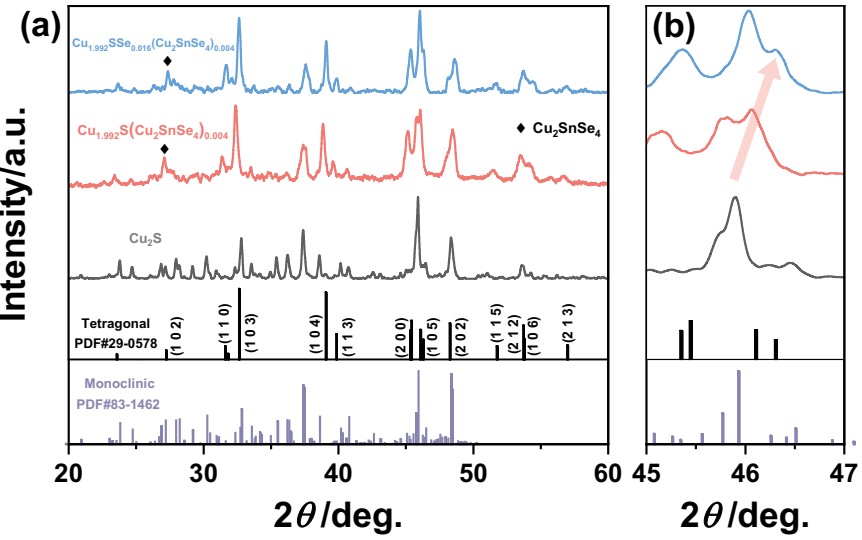

**Fig. 2 | Phase structure characterization of copper sulfide-based materials. a** XRD patterns of the Cu$_2$S, Cu$_{1.992}$S(Cu$_2$SnSe$_4$)$_{0.004}$ and Cu$_{1.992}$SSe$_{0.016}$(Cu$_2$SnSe$_4$)$_{0.004}$ bulk samples. **b** Enlarged XRD patterns of the Cu$_2$S, Cu$_{1.992}$S(Cu$_2$SnSe$_4$)$_{0.004}$ and Cu$_{1.992}$SSe$_{0.016}$(Cu$_2$SnSe$_4$)$_{0.004}$ bulk samples at 45°–47°.

(EPMA) was utilized to observe the distribution of pores and precipitates among the Cu$_{1.992}$SSe$_{0.016}$(Cu$_2$SnSe$_4$)$_{0.004}$ sample, as shown in Supplementary Fig. 3. Energy-dispersive spectroscopy (EDS) spot scanning was performed for the visualized second phase particles with different grain sizes, suggesting that the multiscale precipitates would be Cu$_2$SnSe$_4$. Partial large-scale precipitates are distributed along the grain boundaries, while the nanoprecipitates fill in the nanopores. The special nanostructures of nanoprecipitates embedded in nanopores can usually observed in copper chalcogenides owing to the element emission and the existence of a suitable substrate for crystal growth[41,42]. In addition, the decreased Cu content of the matrix after adding extra Se can be confirmed by the EPMA as well (Supplementary Fig. 4). The existence of these lattice defects is the main reason for the grain refinement in this study, the pores and inclusions can inhibit the migration of grain boundaries.

To further observe the nanoscale lattice defects, TEM was performed for the Cu$_{1.992}$SSe$_{0.016}$(Cu$_2$SnSe$_4$)$_{0.004}$ bulk sample. A high-angle annular dark field (HAADF) image is shown in Fig. 3a, nanopores and nanoprecipitates are easily distinguished. Nanoprecipitates were observed in the TEM image, which consist of Cu, Sn and Se by EDS mapping (Supplementary Fig. 5). Therefore, the introduction of the suitable content of Sn and Se in copper sulfide can produce the Cu$_2$SnSe$_4$ nanoprecipitates. But the precipitates are highly possibly just covered by the matrix without really exposed, and it is too thin comparing with the matrix in the characterization region. It is therefore hard to clearly show the lattice of the Cu$_2$SnSe$_4$ from FFT pattern and/or HRTEM fringe due to the strong effects of the matrix lattices. It can be reasonable proposed that the interfaces between Cu$_2$S and Cu$_2$SnSe$_4$ are incoherent due to their different crystal structures. According to the enlarged TEM image shown in Fig. 3c, there is the region of high-density dislocations, which can be observed through the inverse fast Fourier transform (Fig. 3e). The distribution of a mass of dislocations is consistent with the local concentrated stress by geometric phase analysis (Fig. 3f). The bright parts imply that local concentrated stress exists around the dislocations, which is effective in scattering mid-frequency phonons and restrains the lattice thermal conductivity in the medium temperature range[43,44]. In addition to the noticeable high-density dislocation area, concentrated stress is different from the pristine Cu$_2$S (Supplementary Fig. 6). The lattice stress originates from the structural evolution induced by compositional regulation. First, Cu vacancies are produced in the copper sulfide after introducing a refined ratio of additives, resulting in the formation of

edge dislocations in the matrix and significant lattice distortion[45]. In addition, a mass of interfaces are introduced in the material owing to the grain refinement and the additional nanoprecipitates and nanopores, the lattice stress is also triggered by the dislocations distributed around the interfaces[46]. As shown in Fig. 3g, Cu vacancies in copper sulfide can diffuse and form vacancy clusters during annealing. There are closed rings of edge dislocations when these vacancy clusters further collapse. Higher vacancy concentration would promote the dislocation climbing, resulting in a higher dislocation density. Furthermore, tetragonal and monoclinic phases coexist in the materials near room temperature, which gradually change to hexagonal and then cubic phases with temperature. Obvious atomic mismatch caused by multiphase coexistence leads the strain of compression and extension, and a similar phenomenon is ubiquitous in our samples. In summary, multiscale second phase, refined grains, nanopores and dislocations are introduced in the one-pot modulated copper sulfide-based material, which contribute to scatter heat carriers.

The temperature dependence of the electrical transport properties of the Cu$_2$S, Cu$_{1.992}$S(Cu$_2$SnSe$_4$)$_{0.004}$ and Cu$_{1.992}$SSe$_{0.016}$(Cu$_2$SnSe$_4$)$_{0.004}$ bulk specimens is exhibited in Fig. 4. The electrical conductivity ($\sigma$) gradually increases with temperature and then decreases after undergoing the phase transition temperature (Fig. 4a), indicating that the semiconductor behavior changes to metallic-like behavior. The two turning points of $\sigma$ gradually shift to lower temperatures after introducing refined Sn and Se. Monoclinic Cu$_2$S and tetragonal Cu$_{1.96}$S coexist in the Cu$_{1.992}$S(Cu$_2$SnSe$_4$)$_{0.004}$ and Cu$_{1.992}$SSe$_{0.016}$(Cu$_2$SnSe$_4$)$_{0.004}$ samples, different phase structures affect the turning points of the electrical and thermal transport properties. The increased content of tetragonal Cu$_{1.96}$S results in the enlarged temperature window for achieving high power factor, benefiting for improving average thermoelectric performance. Differential Scanning Calorimetry (DSC) curves of the pristine Cu$_2$S and Cu$_{1.992}$SSe$_{0.016}$(Cu$_2$SnSe$_4$)$_{0.004}$ samples indicate that the introduction of Cu vacancies can decrease the temperature of phase transition for copper sulfides (Supplementary Fig. 7). It is worth noting that the $\sigma$ of Cu$_{1.992}$SSe$_{0.016}$(Cu$_2$SnSe$_4$)$_{0.004}$ is further improved compared to that of the Cu$_{1.992}$S(Cu$_2$SnSe$_4$)$_{0.004}$ sample, which is due to the progressive enhancement in hole concentration by introducing more Cu vacancies. Additionally, the Seebeck coefficient ($S$) of Cu$_{1.992}$S(Cu$_2$SnSe$_4$)$_{0.004}$ sample obviously drops by comparison of that of the pure Cu$_2$S specimen because of the variation in carrier concentration (Fig. 4b). Additionally, $S$ further reduces after introducing

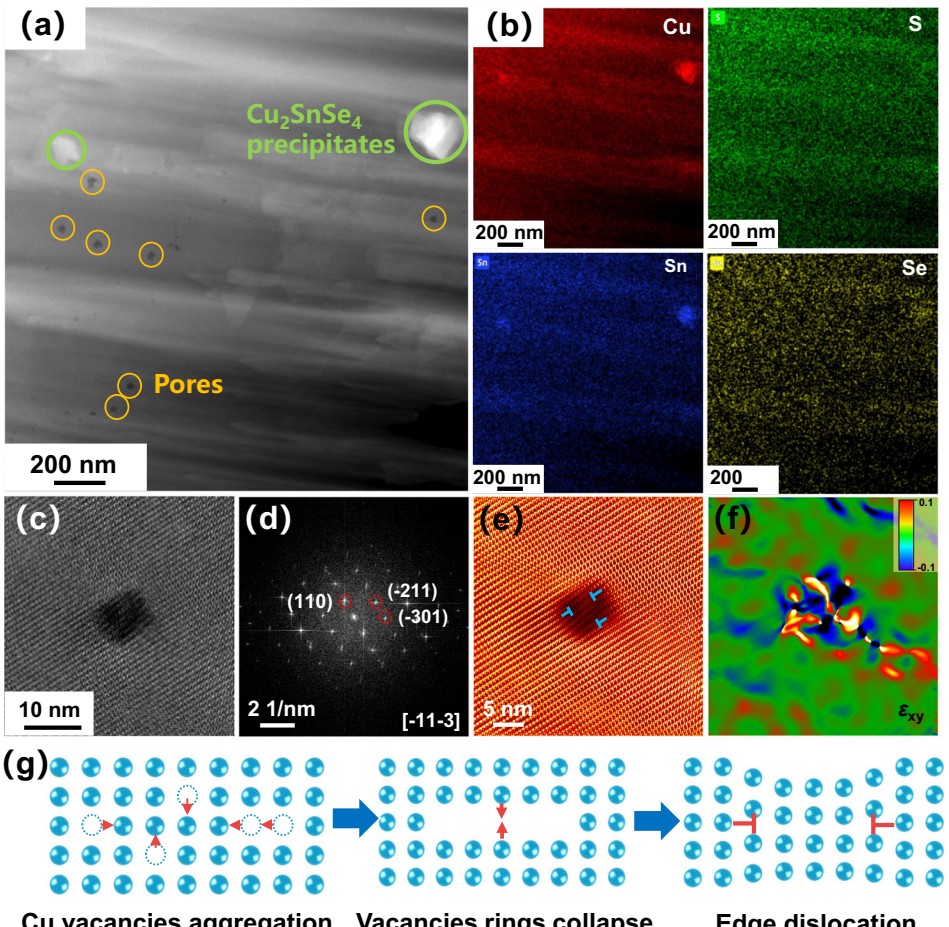

**Cu vacancies aggregation    Vacancies rings collapse    Edge dislocation**

**Fig. 3 | STEM characterization of the Cu$_{1.992}$SSe$_{0.016}$(Cu$_2$SnSe$_4$)$_{0.004}$ bulk specimen. a** HAADF image exhibits the existence of precipitates and nanopores in the material. **b** EDS element mapping of the area in (**a**), indicating that precipitates in the bulk composite are Cu$_2$SnSe$_4$. **c** High-resolution TEM (HRTEM) image of the Cu$_{1.992}$SSe$_{0.016}$(Cu$_2$SnSe$_4$)$_{0.004}$ bulk specimen, expressing the existence of a high-density dislocation area. **d** Corresponding fast Fourier transform (FFT) image of the area, (**e**) the inverse fast Fourier transform (IFFT) image at the selected area. **f** The stress distribution of the whole region in (**c**) by geometric phase analysis (GPA), and the color bar represents −10% to 10% strain. **g** Schematic diagram of the formation of edge dislocation by Cu vacancy.

extra Se, which still remains at a high-level owing to the enhanced carrier effective mass after alloying with Se (Fig. 4e). For the Cu$_{1.992}$SSe$_{0.016}$(Cu$_2$SnSe$_4$)$_{0.004}$ specimen, an $S$ of 196 μVK$^{-1}$ and a $\sigma$ of 377 S cm$^{-1}$ are obtained at 873 K. The electrical transport properties for other Cu$_{2-2x}$SSe$_{4x}$(Cu$_2$SnSe$_4$)$_x$ samples are exhibited in Supplementary Fig. 8. To further investigate the effects of the second phase and Cu vacancies on the charge transport of copper sulfides, Hall measurements and optical band gaps were performed. Figure 4c expresses the carrier concentration ($n_H$) and mobility ($\mu_H$) of the Cu$_2$S, Cu$_{1.992}$S(Cu$_2$SnSe$_4$)$_{0.004}$ and Cu$_{1.992}$SSe$_{0.016}$(Cu$_2$SnSe$_4$)$_{0.004}$ samples at room temperature. As predicted, the $n_H$ reaches 11.09 × 10$^{20}$ cm$^{-3}$ for the Cu$_{1.992}$SSe$_{0.016}$(Cu$_2$SnSe$_4$)$_{0.004}$ sample, which is dramatically higher than that of both pristine Cu$_2$S sample (0.08 × 10$^{20}$ cm$^{-3}$) and Cu$_{1.992}$S(Cu$_2$SnSe$_4$)$_{0.004}$ sample (2.57 × 10$^{20}$ cm$^{-3}$). Herein, introduced Cu vacancies are the main reason for the increase in hole concentration. $\mu_H$ gradually drops from 21.46 cm$^2$V$^{-1}$s$^{-1}$ for Cu$_2$S to 2.03 cm$^2$V$^{-1}$s$^{-1}$ for the Cu$_{1.992}$SSe$_{0.016}$(Cu$_2$SnSe$_4$)$_{0.004}$ specimen. Extra interfaces introduced by grain refinement and pores can deteriorate the carrier transport. The $n_H$ and $\mu_H$ for the Cu$_{1.992}$SSe$_{0.016}$(Cu$_2$SnSe$_4$)$_{0.004}$ sample within a temperature range of 323−873 K were also characterized (Supplementary Fig. 9). Copper sulfides are degenerate semiconductors, thus, there is only a slight variation in carrier concentration for Cu$_{1.992}$SSe$_{0.016}$(Cu$_2$SnSe$_4$)$_{0.004}$ with temperature. Carrier mobility gradually drops with temperature due to the scattering effects of lattice vibration and Cu ion migration. The electronic

structures, total density of states and partial density of states for Cu$_2$S and Cu$_{1.96}$S were calculated based on the optimized crystal structure (Supplementary Fig. 10), implying a typical Dirac cone structure that is in agreement with the previously calculated results[47]. For Cu$_{1.96}$S, Cu vacancies shift the valence band maximum (VBM) upward the conduction band minimum and hence lead to an increased hole concentration. The band shape is almost maintained after introducing Cu vacancies, which is mainly determined by the S sublattice[48]. Furthermore, the optical band gap ($E_g$) for the samples is shown in Fig. 4d, and $E_g$ widens with addition content. Cu vacancies in the copper sulfides can reduce the antibonding character and thus lower the energy position of the VBM[47]. The relationships between the Seebeck coefficient and carrier concentration are fitted by a single parabolic band (SPB) model and depicted in Fig. 4e. The density of states (DOS) effective mass ($m^*$) of the Cu$_{1.992}$SSe$_{0.016}$(Cu$_2$SnSe$_4$)$_{0.004}$ samples is obviously larger than that of pure Cu$_2$S at room temperature, and the $m^*$ of the Cu$_{1.992}$SSe$_{0.016}$(Cu$_2$SnSe$_4$)$_{0.004}$ sample reaches 2.41 $m_O$ at 873 K, which favors maintaining the Seebeck coefficient due to the enhanced DOS near the Fermi level. Owing to the sharp increase in electrical conductivity by inducing Cu vacancies in the material, the power factor ($PF$) for all Cu$_{1.992}$SSe$_{0.016}$(Cu$_2$SnSe$_4$)$_{0.004}$ samples is improved, as shown in Fig. 4f. A peak $PF$ of 1450 μWm$^{-1}$K$^{-2}$ is attained at 873 K for the Cu$_{1.992}$SSe$_{0.016}$(Cu$_2$SnSe$_4$)$_{0.004}$ specimen. More importantly, a higher power factor is realized at a lower temperature for our bulk specimens. Cu vacancies and precipitates induced by one-pot

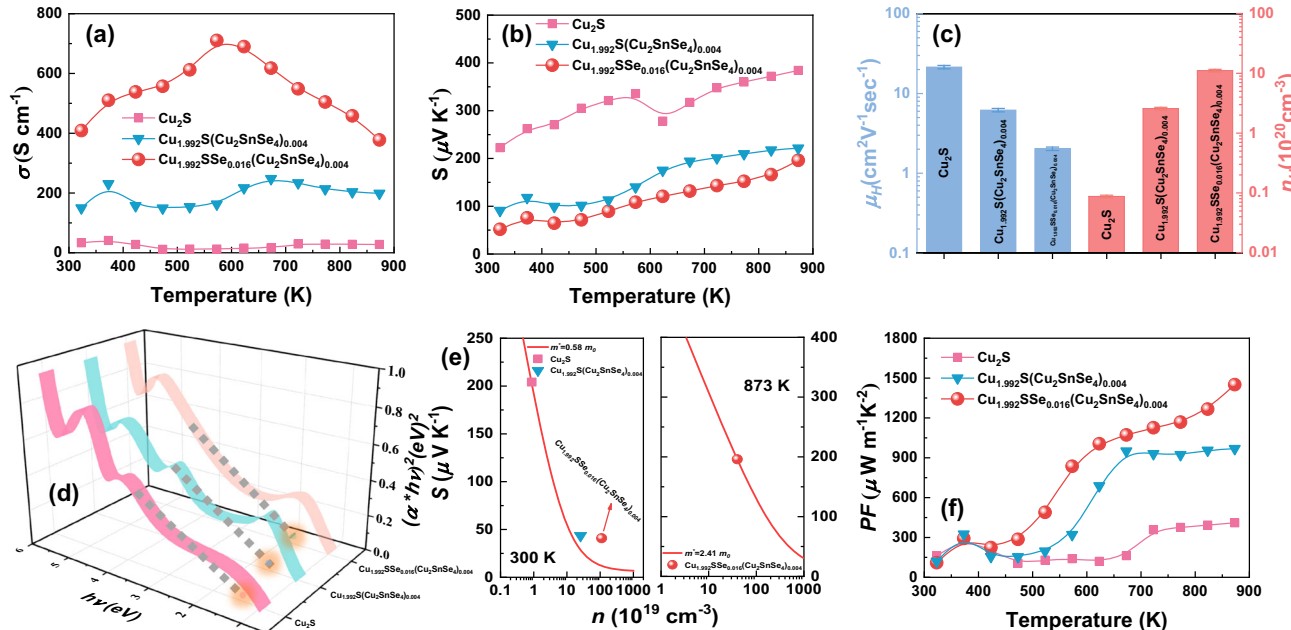

**Fig. 4 | Electrical transport properties of Cu₂S, Cu₁.₉₉₂S(Cu₂SnSe₄)₀.₀₀₄ and Cu₁.₉₉₂SSe₀.₀₁₆(Cu₂SnSe₄)₀.₀₀₄ bulk samples.** Temperature dependence of (**a**) electrical conductivity, (**b**) Seebeck coefficient and (**f**) power factor of Cu₂S, Cu₁.₉₉₂S(Cu₂SnSe₄)₀.₀₀₄ and Cu₁.₉₉₂SSe₀.₀₁₆(Cu₂SnSe₄)₀.₀₀₄ bulk samples. **c** Composition-dependent hall carrier concentration and mobility at 300 K for all bulk samples. **d** $(\alpha h\nu)^2$ vs. $h\nu$ of all specimens, the optical band gap ($E_g$) can be estimated by extrapolating the straight line to $(\alpha h\nu)^2 = 0$. **e** Seebeck coefficient as a function of carrier concentration at 300 K and 873 K. Red lines are obtained by the SPB model and estimating the carrier effective mass, and dots are obtained by experimental data.

modulation facilitate medium-temperature thermoelectric materials to possess larger output power in a wider temperature range.

Figure 5 presents the temperature dependence of the thermal transport properties for Cu₂S, Cu₁.₉₉₂S(Cu₂SnSe₄)₀.₀₀₄ and Cu₁.₉₉₂SSe₀.₀₁₆(Cu₂SnSe₄)₀.₀₀₄ bulk specimens. The total thermal conductivity ($\kappa$) of the Cu₁.₉₉₂SSe₀.₀₁₆(Cu₂SnSe₄)₀.₀₀₄ sample is larger than that of the pristine Cu₂S material over the whole temperature range, as shown in Fig. 5a. $\kappa$ is 0.85 Wm⁻¹K⁻¹ (323 K) and 0.78 Wm⁻¹K⁻¹ (873 K) for Cu₁.₉₉₂SSe₀.₀₁₆(Cu₂SnSe₄)₀.₀₀₄, respectively. To determine the reason for the variation in thermal conductivity, the electronic thermal conductivity ($\kappa_e$) of all specimens is calculated by the Wiedeman-Franz law (Supplementary Fig. 11), and the lattice thermal conductivity is evaluated by subtracting $\kappa_e$ from $\kappa$. The tendency of $\kappa_e$ in the specimens is similar to that of the electrical conductivity, and there is an order of magnitude difference in $\kappa_e$ between the Cu₂S and Cu₁.₉₉₂SSe₀.₀₁₆(Cu₂SnSe₄)₀.₀₀₄ specimens. Therefore, obviously increased carrier concentration and enhanced electrical conductivity are the main reasons for the total thermal conductivity enhancement. Nevertheless, the $\kappa_l$ of Cu₁.₉₉₂SSe₀.₀₁₆(Cu₂SnSe₄)₀.₀₀₄ samples is lower than that of the pristine Cu₂S material, which stems from the existence of various lattice defects in the bulk specimen, contributing to significantly scattering the mid-to-short frequency phonon. As shown in Fig. 5c, the $\kappa_l$ curves for the specimens under different states were modeled to evaluate the effects of multiscale lattice defects on decreasing lattice thermal conductivity by the Debye-Callaway model. Umklapp (U), interface (I), precipitates (P), nanoprecipitates (NP) and dislocation (D) scattering are considered[49,50]. The calculation details, including equations, modulus, grain size, and density of dislocations and precipitates observed by TEM, can be seen in Supplementary Table 2. However, after considering the Umklapp process, various interfaces, multiscale precipitates and dislocation cores and strain, the predicted curve only approximately matches the experimental lattice thermal conductivity of the Cu₁.₉₉₂SSe₀.₀₁₆(Cu₂SnSe₄)₀.₀₀₄ sample at high temperature. The obvious deviation at low temperature might be related to the different crystal structures of copper sulfide-based materials, which would affect the phonon transport, in particular, the

Cu vacancies and the different positions of Cu⁺ ions. This simulation is close to the lattice thermal conductivity calculated by Cahill's theory[51], indicating the great contribution of multiple structural defects to phonon scattering. Because of the strong phonon scattering effect and low-frequency localized vibrations caused by disordered Cu⁺ ion migration and Cu vacancies, the lattice thermal conductivity of copper sulfides is intrinsically low, and the contributions of multiscale lattice defects to lattice thermal conductivity reduction are not as obvious as those in other materials. The frequency dependence of the lattice thermal conductivity of the Cu₁.₉₉₂SSe₀.₀₁₆(Cu₂SnSe₄)₀.₀₀₄ sample is plotted in Supplementary Fig. 12 to better understand the influence of different mechanisms. The Umklapp process, extra grain boundaries and pore interfaces are responsible for scattering low- to medium-frequency phonons for lowering the lattice thermal conductivity at low temperature. The dislocation cores and strain can strongly scatter phonon with mid frequencies to reduce the lattice thermal conductivity at medium temperature, endowing copper sulfides higher thermoelectric performance. In addition, the sound velocities of all specimens were characterized and are exhibited in Fig. 5d. The longitudinal sound speed ($v_l$) gradually increases after adding Sn and Se by one-pot modulation owing to the enhanced bulk density of specimens, but the transverse sound velocity ($v_t$) decreases. Se-alloying has been proved to be effective in reducing bonding energy of the Cu₂S materials, resulting in the decreased speed of sound and therefore the lower lattice thermal conductivity[37]. Ultimately, although the increased hole concentration and electrical conductivity lead to enhanced carrier thermal conductivity, highly strengthened phonon scattering by various mechanisms assists the ultralow lattice thermal conductivity and the maintained total thermal conductivity.

Benefiting from the significantly enhanced electrical transport properties and the maintained thermal conductivity, an overall improved thermoelectric figure of merit ($ZT$) is realized for the one-pot modulated samples (Supplementary Fig. 13). A peak $ZT$ beyond 1.6 is achieved for the Cu₁.₉₉₂SSe₀.₀₁₆(Cu₂SnSe₄)₀.₀₀₄ sample at 873 K (Fig. 6a). According to the experimental carrier concentration and mobility and lattice thermal conductivity at different temperatures,

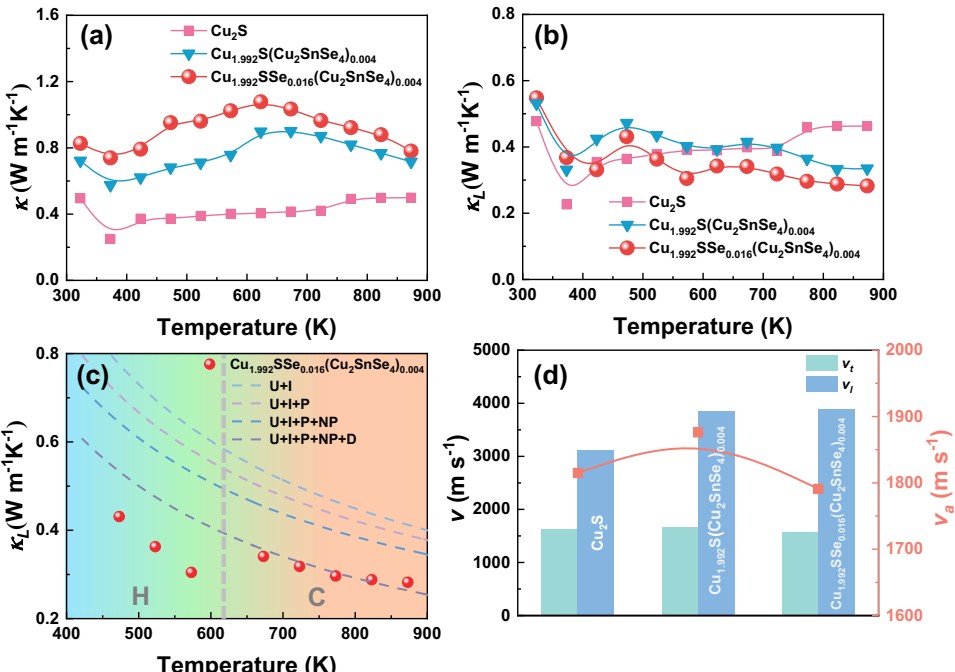

**Fig. 5 | Thermal transport properties of copper sulfide-based bulk composites.** Temperature dependence of (**a**) thermal conductivity and (**b**) lattice thermal conductivity for $Cu_2S$, $Cu_{1.992}S(Cu_2SnSe_4)_{0.004}$ and $Cu_{1.992}SSe_{0.016}(Cu_2SnSe_4)_{0.004}$ bulk samples. **c** Fitting lattice thermal conductivity by the Callaway thermal conductivity model. **d** Sound velocity of $Cu_2S$, $Cu_{1.992}S(Cu_2SnSe_4)_{0.004}$ and $Cu_{1.992}SSe_{0.016}(Cu_2SnSe_4)_{0.004}$ bulk samples.

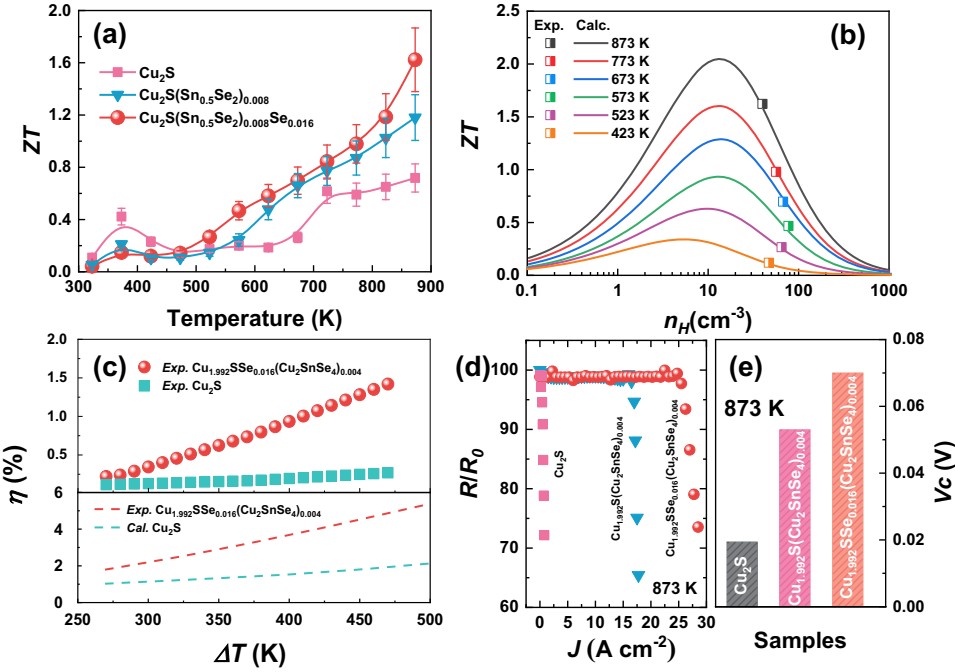

**Fig. 6 | Thermoelectric performance of copper sulfide-based bulk composites.** Temperature dependence of (**a**) dimensionless figure of merit *ZT* of $Cu_2S$, $Cu_{1.992}S(Cu_2SnSe_4)_{0.004}$ and $Cu_{1.992}SSe_{0.016}(Cu_2SnSe_4)_{0.004}$ bulk samples. **b** Carrier concentration dependence of the *ZT* value at different temperatures by using the SPB model. **c** Experimental and estimated conversion efficiency as a function of temperature gradient for the $Cu_{1.992}SSe_{0.016}(Cu_2SnSe_4)_{0.004}$ sample. **d** Related electrical resistivity ($R/R_0$) of the $Cu_{1.992}SSe_{0.016}(Cu_2SnSe_4)_{0.004}$, $Cu_{1.96}S$ and $Cu_2S$ samples with increased current density at 873 K. **e** The critical voltage ($V_c$) of the pristine $Cu_2S$ and that with Sn and Se addition at 873 K.

the *ZT* values of the $Cu_{1.992}SSe_{0.016}(Cu_2SnSe_4)_{0.004}$ sample are expressed as a function of carrier concentration in Fig. 6b, which are located at the theoretical lines simulated by the SPB model. Nevertheless, the excessive carrier concentration caused by the induced Cu vacancies results in the actual *ZT* being lower than the theoretical value. Therefore, the $Cu_{1.992}SSe_{0.016}(Cu_2SnSe_4)_{0.004}$ material still has great potential to achieve higher *ZT* by further introducing n-type dopants or decreasing the bonding energy. The temperature gradient ($\Delta T$) dependence of the conversion efficiency of the $Cu_{1.992}SSe_{0.016}(Cu_2SnSe_4)_{0.004}$ sample is calculated ($\eta_{cal}$) after

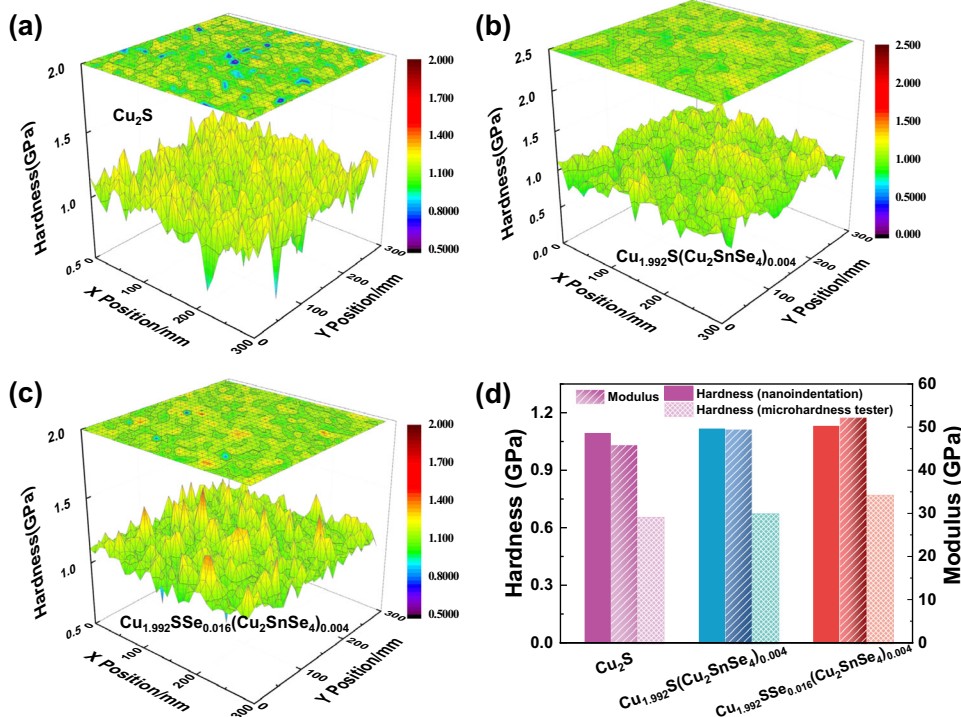

**Fig. 7 | Mechanical performance of copper sulfide-based bulk samples.** 3D cloud diagram of hardness for (**a**) pure $Cu_2S$, (**b**) $Cu_{1.992}S(Cu_2SnSe_4)_{0.004}$ and (**c**) $Cu_{1.992}SSe_{0.016}(Cu_2SnSe_4)_{0.004}$ bulk samples. **d** Average modulus and average hardness of $Cu_2S$, $Cu_{1.992}S(Cu_2SnSe_4)_{0.004}$ and $Cu_{1.992}SSe_{0.016}(Cu_2SnSe_4)_{0.004}$ samples measured by using a nanoindentation instrument and microhardness tester, respectively.

considering the Thomson heat, as shown in Fig. 6c[52]. With a cold-side temperature of 300 K, the $Cu_{1.992}SSe_{0.016}(Cu_2SnSe_4)_{0.004}$ sample yields an $\eta_{cal}$ greater than 5% when $\Delta T$ reaches 470 K, which is approximately three times higher than that of the pristine $Cu_2S$ sample. The power generation efficiency ($\eta_{exp}$) of the fabricated single-leg thermoelectric modules by $Cu_{1.992}SSe_{0.016}(Cu_2SnSe_4)_{0.004}$ material is measured by mini-PEM, as shown in Fig. 6c. The current-dependent voltage ($V$), output power ($P$), heat flux ($Q$) and conversion efficiency at different hot-side temperatures for the single-leg TE module are shown in Supplementary Fig. 14. Because of the contact resistance between the electrode and TE module, the measured efficiency is overall lower than the evaluated efficiency. A $\eta_{exp}$ of 1.5% is obtained at a temperature difference of 470 K, which is still higher than that of the module consisting of pure $Cu_2S$ because of the enhanced average $ZT$.

Electrical stability is also important in practical device applications and can be evaluated by the relative resistance ($R/R_O$) of the $Cu_{1.992}SSe_{0.016}(Cu_2SnSe_4)_{0.004}$ and $Cu_{1.992}S(Cu_2SnSe_4)_{0.004}$ samples under increased current density at 873 K by comparison of the pure $Cu_2S$ material. The $R/R_O$ for the pristine $Cu_2S$ material dramatically decreases after starting subjecting a small current density, indicating poor electrical stability and material failure under an electrical field (Fig. 6d). The critical current density ($J_c$) indicates the strength of the external field that enables the concentration of $Cu^+$ ions in the Cu-based superionic conductors to meet the chemical potential for metallic Cu deposition. A higher $J_c$ is obtained for the $Cu_{1.992}S(Cu_2SnSe_4)_{0.004}$ material compared with the $Cu_2S$ sample under the same conditions owing to the inhabitation of long-range migration of Cu ion by the precipitates. Additionally, $R/R_O$ for the $Cu_{1.992}SSe_{0.016}(Cu_2SnSe_4)_{0.004}$ sample drops until the current density reaches approximately 25 A cm$^{-2}$, suggesting that the introduced Cu vacancies by adding extra Se in the copper sulfides are beneficial for decreasing the concentration of Cu ions and optimizing the stability of copper sulfides. Furthermore, in order to more reasonably evaluate the electrical stability of the superionic conductors, the critical voltage ($V_c$) of the

$Cu_{1.992}SSe_{0.016}(Cu_2SnSe_4)_{0.004}$ and $Cu_{1.992}S(Cu_2SnSe_4)_{0.004}$ samples at 873 K are shown in Fig. 6e, the calculated details can be seen in the reported works[53]. $V_c$ of the pristine $Cu_2S$ is slightly lower than the reported result due to the higher measurement temperature. Besides, higher $V_c$ of $Cu_{1.992}SSe_{0.016}(Cu_2SnSe_4)_{0.004}$ indicates that the spontaneously generated Cu vacancies and multiscale precipitates contribute to improve the electrical stability of copper sulfides. Owing to the introduction of Cu vacancies and precipitates in the copper sulfides, the thermoelectric performance of the Sn and Se-added samples is maintained in the cycling test (Supplementary Fig. 15).

The mechanical performance of all bulk specimens was characterized by nanoindentation with the Nano-Blitz 3D method (Supplementary Fig. 16). 3D diagrams of hardness ($H$) (Fig. 7a–c) suggest that $H$ for the $Cu_{1.992}SSe_{0.016}(Cu_2SnSe_4)_{0.004}$ sample is obviously higher than that of both the $Cu_2S$ and $Cu_{1.992}S(Cu_2SnSe_4)_{0.004}$ bulk samples. The average hardness and modulus of the bulk composites are plotted in Fig. 7d. Gradually enhanced mechanical properties are ascribed to grain refinement and dispersion strengthening. A large number of grain boundaries and second phase interfaces are introduced by one-pot modulation, which are capable of blocking the dislocation motion and enhancing the hardness of composites. The Vickers hardness of the composites is also characterized by a microhardness tester (Fig. 7d). The similar tendency of microhardness but lower values compared with $H$ measured by nanoindentation (Fig. 7d and Supplementary Fig. 17) reflect those widespread pores in the $Cu_{1.992}SSe_{0.016}(Cu_2SnSe_4)_{0.004}$ samples weaken the overall hardness of the materials. The enhanced mechanical performance of composites by structural and compositional evolution would promote copper sulfides further engaging as p-type legs in thermoelectric device assembly.

In summary, XRD, FESEM, EMPA, EDS and TEM results demonstrate that employing one-pot modulation to fabricate copper sulfide-based materials can introduce an in-situ generated precipitates while effectively tuning the Cu content for synergistically improving thermal

and electrical properties. Multiscale precipitates, nanopores and dislocations are produced in the specimens, an ultralow lattice thermal conductivity is obtained through strengthening the phonon scattering effect by various mechanisms. Meanwhile, the introduced Cu vacancies have significant potential for improving electrical conductivity while reducing the phase transition temperature, and an overall and obviously improved power factor is realized. Ultimately, a peak $ZT$ of 1.6 is obtained for the $Cu_{1.992}SSe_{0.016}(Cu_2SnSe_4)_{0.004}$ bulk sample at 873 K. The introduced precipitates and Cu vacancies can also suppress Cu ion migration and improve the electrical stability of Cu-based superionic conductors. Therefore, this one-pot modulation promotes copper sulfide-based composites as potential candidates for thermoelectric applications, which is worth facilitating in other material systems.

## Methods
### Synthesis
$Cu_2S$, $Cu_{1.992}S(Cu_2SnSe_4)_{0.004}$ and $Cu_{2-2x}SSe_{4x}(Cu_2SnSe_4)_x$ ($x = 0$, 0.003, 0.0035, 0.004, 0.005 and 0.006) materials were fabricated by high-temperature melting, long-time annealing, high-energy ball milling and spark plasma sintering. High-purity raw elements, Cu (pellet bulk, 99.999%), S (pellet, 99.999%), Sn (pellet, 99.999%) and Se (pellet, 99.999%), were weighed out according to the stoichiometric proportions, placed in a quartz tube, and then the tube was evacuated at $10^{-4}$ Pa and sealed. The samples were heated to 1373 K in 18 h, keeping at 1373 K for 12 h, and then cooled to 1073 K within 24 h. The annealing process is holding at 1073 K for 7 days and then cooling to 300 K by shutting down the furnace power. The obtained ingots were grinded into fine powders under a protective atmosphere condition (95 vol% Ar) by ball-milling (Retsch Emax, German) for 30 min with a speed of 800 rpm. The prepared powders were put into a graphite mold with the size of φ20 mm, which was sintered at 773 K under 50 MPa pressure in 5 min by using the spark plasma sintering system (Sumitomo SPS1050, Japan). The bulk samples were subsequently cut and polished.

### Characterization
X-ray diffraction (XRD MiniFlex600 Rigaku, Japan) was utilized to detect the phase structure of the prepared bulk samples from diffraction angles from 20° to 60° at a speed of 5°/min by using Cu Kα radiation ($\lambda = 1.54$ Å). The morphologies of the fractured bulk samples were inspected using electron probe microanalysis (EPMA, JEOL, JXA-8230, Japan) and field emission scanning electron microscopy (FESEM, ZEISS, Sigma 300, Germany) equipped with energy dispersive spectroscopy (EDS, JED-2300T). Scanning transmission electron microscopy (STEM, FEI Titan) was utilized to investigate the microstructures and nanoprecipitates in the bulk samples. The electrical conductivity ($\sigma$) and the Seebeck coefficient ($S$) were simultaneously measured by a resistivity and Seebeck coefficient measurement system (ZEM-3, Advance Riko, Japan) in a low-pressure helium gas environment. Additionally, the thermal conductivity ($\kappa$) of the bulk composites was calculated by $\kappa = D \times C_p \times \rho$, where the thermal diffusivity $D$ was measured using an LFA457 (Netzsch, Germany) laser flash apparatus, the specific heat ($C_p$) was calculated using $C_p = 3Nk_B$, and the mass density $\rho$ was measured by the Archimedes method. The DSC curves of the samples in the temperature range of 310–773 K are obtained by the Netzsch STA 449 F3 (Germany) instrument under flowing Ar gas, and the heating speed is 10 K min$^{-1}$. The Hall carrier concentration ($n_H$) and mobility ($\mu_H$) of the thin samples were tested by using a Hall effect measurement system (Ecopia, HMS-7000, Korea). The microhardness ($H$) and Young's modulus ($E$) of all bulk samples were measured by utilizing a nanoindenter, 900 points were presses on the polished surface of samples, and a target load of 20 mN and a feature height of 2000 nm were set (iMicro KLA., USA). The relative resistivity of the $Cu_{1.992}SSe_{0.016}(Cu_2SnSe_4)_{0.004}$ bulk sample was measured by a homemade instrument including a vacuum furnace and an electrochemical station. The absolute error of all experimental data can be assumed to be 3%–4%, so the error bar of $ZT$ would be 15%–20%.

## Data availability
The data that support the findings of this study are available from the corresponding author on request.

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

## Acknowledgements

We thank Dr. Lei Jin from Forschungszentrum Jülich for the help of TEM analysis. This work was supported by the Outstanding Youth Fund of Yunnan Province (Grant No. 202201AV070005), National Key R&D Program of China (No. 2022YFF0503804), Yunnan Provincial Natural Science Key Fund (Grant No. 202101AS070015), National Natural Science Foundation of China (Grant No. 52162029), Yunnan Major Scientific and Technological Projects (grant NO. 202302AG050010). All authors have given approval to the final version of the manuscript.

## Author contributions

Z.H.G. and J.F. conceived the project; Y.X.Z. prepared the materials; Y.X.Z., X.Y., C.Y.W., T.Y.Y., Y.C.S. and Z.Y.W. tested TE and mechanical performance; Q.Y.H. and Q.S. performed the TEM characterization; Y.X.Z. analyzed the experimental data; Y.X.Z. wrote the manuscript. All authors participated in the data analysis and the manuscript editing.

## Competing interests

The authors declare no competing interests.
