## [Peer Review File · Nature Communications]

Synergistically optimized electron and phonon transport in high-performance copper sulfides thermoelectric materials via one-pot modulationREVIEWER COMMENTS

Reviewer #1 (Remarks to the Author):

Hereby, to decouple carrier/phonon transport and achieve high thermoelectric performance in copper sulfide materials, a one-pot approach realizing simultaneous second phase and Cu vacancies modulation is proposed. Multiple lattice defects, including in-situ precipitates, dislocations, and nanopores are produced by adding a refined ratio of Sn and Se. An ultralow lattice thermal conductivity is therefore obtained. Additionally, extra Se is added in the copper sulfide for optimizing electrical transport properties by inducing generating Cu vacancies. Ultimately, an excellent figure of merit of ~ 1.6 at 873 K is realized in the $\text{Cu}_2\text{S}(\text{Sn}_{0.5}\text{Se}_2)_{0.008}\text{Se}_{0.016}$ bulk sample.

Overall speaking, this study is comprehensive and worth publication. However, there are some severe drawbacks make the current version can be hardly accepted for publication. As a brief summary, the experiment design logic is hard to understand. The structure characterization is not trustable. The relationship between performance and structure is not clear. What are exactly decoupled and how it is decoupled are also not straight-forward and clear enough. More detailed evidence, expression and discussions are necessary. More detailed comments are as below:

1. At the beginning of the Abstract, the 'coupling of carrier, phonon and ion transport', which is different from the title highlighted 'electron and phonon transport'. Please think carefully about the key highlight of this study, re-consider corresponding descriptions and avoid misunderstanding or confusion of readers.

2. As the first step of this study, additional Sn and Se have been simultaneously introduced in the sample $\text{Cu}_2\text{S}(\text{Sn}_{0.5}\text{Se}_2)_{0.008}$. However, the design logic is missing. Why the composition is designed like this? If the purpose is introducing Cu_2SnSe_4 second phase as proposed in Fig. 1, why there is no design of additional Cu?

3. Additionally, assuming all additional $(\text{Sn}_{0.5}\text{Se}_2)_{0.008}$ can form Cu_2SnSe_4 second phase, the nominal composition can be re-written into $\text{Cu}_{1.92}\text{S}(\text{CuSn}_{0.5}\text{Se}_2)_{0.008}$, where 4% Cu vacancies can be expected with the Cu stoichiometry of 1.92. However, other than XRD, no other direct evidence of compositional characterization has been seen, which can hardly support the claimed composition change. Please consider supplementing some EPMA analyses.

4.1. The claimed Cu_2SnSe_4 peak in XRD pattern is overlapping with (102) of Tetragonal Cu_2S , as well as another peak of Monoclinic $\text{Cu}_{1.96}\text{S}$, which can be hardly used as strong evidence of Cu_2SnSe_4 second phase. More evidence of Cu_2SnSe_4 second phase should be provided, such as more accurate TEM-EDS analysis, or some HRTEM analysis. Considering Cu_2SnSe_4 should be a cubic phase with the $F\bar{4}3m$ space group, and lattice parameter of ~ 5.7 Å. It should be obviously different from other phases, and can be identified by TEM.

4.2. Meanwhile, the TEM analysis here demonstrates exactly the SAME lattice between Cu_2SnSe_4 and Cu_2S matrix. I can hardly believe this is correct, as they should be different phases, where the Cu_2SnSe_4 should be cubic and the Cu_2S should be either monoclinic or Tetragonal. This result is inconsistent with known structure of Cu_2SnSe_4 . This is highly possibly just a misunderstanding, where there is another thick layer of Cu_2S covering the Cu_2SnSe_4 precipitates, presenting the Cu_2S lattice. If further polished, Cu_2SnSe_4 should be visible, or move to thinner areas for characterization.

5. While multiple factors are changing simultaneously, including structure (monoclinic/tetragonal), composition (Cu vacancies), precipitates, and nanopores, how did the authors understand which is the key factor influencing the performance? It would be much more trustworthy if the authors could realize better control of these factors. At least,

more detailed discussions about the influence of these factors should be provided.

6. Limited by technique accuracy, employing room-temperature Hall data for carrier transport analysis is understandable. However, the temperature inconsistency should at least be stated.

7. From the performance data, I can hardly see any result supporting the claim of 'decoupled carrier phonon transport'. Although, a low lattice thermal conductivity is approached, the high electrical performance is mainly due to high carrier concentration. Instead, with step-by-step optimization, the carrier mobility are continuously decreasing. Please think more carefully about term-use, and avoid misuse, or provide straight-forward evidence/results.

Reviewer #2 (Remarks to the Author):

This manuscript describes the thermoelectric properties of p-type Cu₂S. The authors claim that a method to simultaneously achieve precipitation and Cu vacancy modulation is proposed, effectively decoupling electron and phonon transport in copper sulfide, and achieving a high ZT of 1.6 at 873 K. However, the effects of precipitates and copper vacancies on the thermoelectric properties of Cu₂S have been published in Nano Energy 49, 267-273 (2018) and J Mater. Sci-Mater. El. 30, 5177-5184 (2019), and the ZT of Cu₂S has exceeded 2.1 by controlling the microstructure (Angew. Chem. Int. Ed., 2022, 61.45: e202212885.). In addition, some of the experimental observations need clear explanations, as shown below:

1. The description of the increase in Cu vacancies caused by excess Se requires more experimental or theoretical research, since Cu vacancies play an important role in the thermoelectric and microstructure of Cu₂S in this work.
2. The composition of the precipitate needs double check. Only one Bragg peak shown in Fig. 2, it is hard to index Cu₂SnSe₄. Also, the XRD patterns of Cu₂SnSe₄ and Cu₂SnSe₃ are nearly the same. Thus, more study is needed to verify the precipitates.
3. Fig. 3, the nanoprecipitates shown in C is not the one shown in A, thus a detailed analysis of the precipitate shown in C is needed. The crystal structure of Cu₂S and Cu₂SnSe₄ or Cu₂SnSe₃ is largely different, it is hard to get a coherent boundary between them.
4. Fig. 4, the electrical conductivity and carrier concentration of Cu₂S(SnSe)Se is much higher than that of Cu₂S(SnSe), while the Seebeck coefficient of these two samples are nearly the same. This is strange in thermoelectric materials; the authors need to explain it clearly.
5. The authors should provide thermoelectric data for at least 3 heating-cooling cycles to point out the stability of this material.

Reviewer #3 (Remarks to the Author):

In this article, the authors fabricated a series of copper sulfide materials with Cu vacancies and various defects by incorporating Cu₂S with Sn and Se. The high-density dislocations, multiscaled precipitates and pores were simultaneously obtained in the bulk material and the effects of various mechanisms on scattering mid-to-low frequency phonon were discussed. They found the carrier concentration and electrical properties of the copper sulfides were effectively adjusted by spontaneously tuned Cu content. This work provides an interesting method to improve the thermoelectric performance and stability of copper sulfides. My comments/suggestions about this work are listed below.

- (1) The stress distribution in Fig. 3d is interesting. Have the authors performed the geometric phase analysis on the pristine Cu₂S?
- (2) In Fig. 4a, the turning points of electrical conductivity gradually shift to the lower temperature with increasing the Sn and Se content. The authors should clarify the reason.
- (3) The mechanism for the formation of nanopores is a little obscure? More discussions are required.
- (4) What about the thermal stability of the Sn and Se-added samples? The cycling test should be performed.
- (5) In Figure 5, the shear sound velocity gradually decreases with Sn and Se content, the authors should explain the reason.
- (6) In Figure 3c, two special nanostructures are marked by “g1” and “g2”, respectively. However, there are no corresponding figure numbers.
- (7) Have the authors measured the DSC of the Sn and Se-added samples? Will adding Sn and Se change the phase transition behavior?
- (8) What about the uncertainty of the experimental data in this study? The error bars should be added in the ZT results.
- (9) In Fig. 6d, the authors evaluated the stability of their samples by measuring the relative resistance variation under different current density. Actually, the critical voltage is a more intrinsic parameter to evaluate the stability of superionic conductors. More details can be found in NATURE COMMUNICATIONS (2018) 9:2910.

REVIEWER COMMENTS

Reviewer #1 (Remarks to the Author):

Hereby, to decouple carrier/phonon transport and achieve high thermoelectric performance in copper sulfide materials, a one-pot approach realizing simultaneous second phase and Cu vacancies modulation is proposed. Multiple lattice defects, including in-situ precipitates, dislocations, and nanopores are produced by adding a refined ratio of Sn and Se. An ultralow lattice thermal conductivity is therefore obtained. Additionally, extra Se is added in the copper sulfide for optimizing electrical transport properties by inducing generating Cu vacancies. Ultimately, an excellent figure of merit of ~ 1.6 at 873 K is realized in the $\text{Cu}_2\text{S}(\text{Sn}_{0.5}\text{Se}_2)_{0.008}\text{Se}_{0.016}$ bulk sample.

Overall speaking, this study is comprehensive and worth publication. However, there are some severe drawbacks make the current version can be hardly accepted for publication. As a brief summary, the experiment design logic is hard to understand. The structure characterization is not trustable. The relationship between performance and structure is not clear. What are exactly decoupled and how it is decoupled are also not straight-forward and clear enough. More detailed evidence, expression and discussions are necessary. More detailed comments are as below:

Response: We thank the referee 1 for his/her comments and suggestions. And we have tried our best to revise the manuscript. The experiment design logic of this study is clarified, more detailed evidence, expression and discussions are added as well.

1. At the beginning of the Abstract, the ‘coupling of carrier, phonon and ion transport’, which is different from the title highlighted ‘electron and phonon transport’. Please think carefully about the key highlight of this study, re-consider corresponding descriptions and avoid misunderstanding or confusion of readers.

Response: Thanks for your valuable suggestion. The key highlight of this study is synergistically optimized electron and phonon transport in the high-performance copper sulfides thermoelectric materials. The related descriptions have been revised in the whole manuscript. In the abstract part: the description has been changed to “Optimizing thermoelectric conversion efficiency requires the compromise of electrical and thermal properties of materials, which are hard to simultaneously improve due to the strong coupling of carrier and phonon transport. Herein, a one-pot approach realizing simultaneous second phase and Cu vacancies modulation is proposed, which is effective in synergistically optimizing thermoelectric performance in copper sulfides.”

2. As the first step of this study, additional Sn and Se have been simultaneously introduced in the sample $\text{Cu}_2\text{S}(\text{Sn}_{0.5}\text{Se}_2)_{0.008}$. However, the design logic is missing. Why the composition is designed like this? If the purpose is introducing Cu_2SnSe_4 second phase as proposed in Fig. 1, why there is no design of additional Cu?

Response: Thanks for your questions. The strategies for synergistically optimizing the thermoelectric properties of copper sulfide materials of this study are shown in Fig. 1. The first step aims at spontaneously generating Cu_2SnSe_4 second phase by adding the refined ratio of Sn and Se, this process consumes with a tiny of Cu of the Cu_2S matrix. $\text{Cu}_2\text{S}(\text{Sn}_{0.5}\text{Se}_2)_{0.008}$ can be rewrite as $\text{Cu}_{1.992}\text{S}(\text{Cu}_2\text{SnSe}_4)_{0.004}$. In the Cu_{2-x}S system, it is well-known that the Cu vacancies are beneficial to increase the electrical conductivity. [*Adv. Mater.* 26, 3974 (2014)] The second step aims at introducing more Cu vacancies by adding extra Se, to further increase the carrier concentration for enhancing the electrical transport properties. The content of $\text{Cu}_{1.992}\text{SSe}_{0.016}(\text{Cu}_2\text{SnSe}_4)_{0.004}$ is therefore designed, the results showed that the extra Se introduced more Cu vacancies for effectively improving the electrical conductivity of the sample. The content of Cu vacancies generated by the second step is obviously

higher than the first one. Therefore, the effect of Cu deficiency caused by the first step is covered by the second step, and there is no design of additional Cu in the first step.

Fig. 1 The designed one-pot strategy of utilizing second phase and Cu vacancies modulation to optimizing thermoelectric properties of copper sulfides. (a) The process of a one-pot modulation strategy in adjusting electrical and thermal transport properties. (b) Lattice thermal, (c) average power factor and (d) ZT of Cu_2S , $\text{Cu}_{1.992}\text{S}(\text{Cu}_2\text{SnSe}_4)_{0.004}$ and $\text{Cu}_{1.992}\text{SSe}_{0.016}(\text{Cu}_2\text{SnSe}_4)_{0.004}$ bulk samples. Peak ZT values at 873 K for other copper-based thermoelectric materials are added for comparison, and the data were taken from ref. ^{18,23,24-37}.

3. Additionally, assuming all additional $(\text{Sn}_{0.5}\text{Se}_2)_{0.008}$ can form Cu_2SnSe_4 second phase, the nominal composition can be re-written into $\text{Cu}_{1.92}\text{S}(\text{CuSn}_{0.5}\text{Se}_2)_{0.008}$, where 4% Cu vacancies can be expected with the Cu stoichiometry of 1.92. However, other than XRD, no other direct evidence of compositional characterization has been seen, which can hardly support the claimed composition change. Please consider supplementing some EPMA analyses.

Response: Thanks for your valuable suggestion. We agree that the nominal composition of $\text{Cu}_2\text{S}(\text{Sn}_{0.5}\text{Se}_2)_{0.008}$ can be rewrite as $\text{Cu}_{1.992}\text{S}(\text{Cu}_2\text{SnSe}_4)_{0.004}$, the Cu stoichiometry is 1.992 instead of 1.92. There are only 0.8% Cu vacancies in the $\text{Cu}_{1.992}\text{S}(\text{Cu}_2\text{SnSe}_4)_{0.004}$ sample. Similarly, the

$\text{Cu}_2\text{S}(\text{Sn}_{0.5}\text{Se}_2)_{0.008}\text{Se}_{0.016}$ can be rewrite as $\text{Cu}_{1.992}\text{SSe}_{0.016}(\text{Cu}_2\text{SnSe}_4)_{0.004}$, additional Cu vacancies are introduced in this specimen. In order to characterize the effect of additional Se on the compositional evolution of the material, EPMA analyses were performed for the $\text{Cu}_{1.992}\text{S}(\text{Cu}_2\text{SnSe}_4)_{0.004}$ and $\text{Cu}_{1.992}\text{SSe}_{0.016}(\text{Cu}_2\text{SnSe}_4)_{0.004}$ samples, respectively (Supplementary Fig. 5). The ratio of Cu and S/Se in the $\text{Cu}_{1.992}\text{S}(\text{Cu}_2\text{SnSe}_4)_{0.004}$ sample is close to the pristine Cu_2S , indicating that the introduction of the designed content of Sn and Se in the Cu_2S would not obviously change the stoichiometric ratio of main phase. Furthermore, the introduction of extra Se without changing other condition can significantly decrease the Cu content of the matrix, since that the generation of Cu-Se-S solid solution consumes with Cu of the raw materials. Therefore, the ratio of Cu and S/Se for the $\text{Cu}_{1.992}\text{SSe}_{0.016}(\text{Cu}_2\text{SnSe}_4)_{0.004}$ sample reduces to about 1.96. Related results have been added in the supplementary information as Supplementary Fig. 5, and the related descriptions have been added in the supplementary information as well.

Supplementary Fig. 5 EPMA results of the typical samples. Backscattered electron (BSE) images of the (a) $\text{Cu}_{1.992}\text{S}(\text{Cu}_2\text{SnSe}_4)_{0.004}$ and (b) $\text{Cu}_{1.992}\text{SSe}_{0.016}(\text{Cu}_2\text{SnSe}_4)_{0.004}$ sample, (c) the content of

various elements in these two specimens by EPMA.

4.1. The claimed Cu_2SnSe_4 peak in XRD pattern is overlapping with (102) of Tetragonal Cu_2S , as well as another peak of Monoclinic $\text{Cu}_{1.96}\text{S}$, which can be hardly used as strong evidence of Cu_2SnSe_4 second phase. More evidence of Cu_2SnSe_4 second phase should be provided, such as more accurate TEM-EDS analysis, or some HRTEM analysis. Considering Cu_2SnSe_4 should be a cubic phase with the $F\bar{4}3m$ space group, and lattice parameter of ~ 5.7 Å. It should be obviously different from other phases, and can be identified by TEM.

Response: Thanks for your valuable suggestion. We agree that it is hard to prove the existence of Cu_2SnSe_4 in the samples only by XRD due to the overlap of the diffraction peaks. Therefore, TEM of the $\text{Cu}_{1.992}\text{SSe}_{0.016}(\text{Cu}_2\text{SnSe}_4)_{0.004}$ sample is performed. Nanoprecipitates exist in the sample, which can be confirmed as the Cu-Sn-Se compound by EDS mapping (Supplementary Fig. 7 (c-f)). Additionally, the HRTEM of the nanoprecipitate and the corresponding SAED are shown in Supplementary Fig. 7 (b). The lattice spacing of 0.21 nm is consistent with the (220) plane of cubic Cu_2SnSe_4 . Therefore, the introduction of the suitable content of Sn and Se in copper sulfide can produce the Cu_2SnSe_4 nanoprecipitates. The related results have been added in the supplementary information as Supplementary Fig. 7, and the corresponding discussions have been added in the revised manuscript.

Supplementary Fig. 7 TEM characterization for the typical precipitate. (a) HAADF image of $\text{Cu}_{1.992}\text{SSe}_{0.016}(\text{Cu}_2\text{SnSe}_4)_{0.004}$ sample, showing the existence of nanoprecipitates. (b) HRTEM of the precipitate with the inset of the corresponding SAED. EDS element mapping of the area in (a), (c) is Cu, (d) is Sn, (e) is Se and (f) is S.

4.2. Meanwhile, the TEM analysis here demonstrates exactly the SAME lattice between Cu_2SnSe_4 and Cu_2S matrix. I can hardly believe this is correct, as they should be different phases, where the Cu_2SnSe_4 should be cubic and the Cu_2S should be either monoclinic or Tetragonal. This result is inconsistent with known structure of Cu_2SnSe_4 . This is highly possibly just a misunderstanding, where there is another thick layer of Cu_2S covering the Cu_2SnSe_4 precipitates, presenting the Cu_2S lattice. If further polished, Cu_2SnSe_4 should be visible, or move to thinner areas for characterization.

Response: We are sorry for lacking the detailed evidence and explanation in the previous manuscript that makes you confuse. The coherent interfaces exist between the copper sulfide matrix and the Cu-Se-S compounds. This type of nanoprecipitates can be observed elsewhere in the $\text{Cu}_{1.992}\text{SSe}_{0.016}(\text{Cu}_2\text{SnSe}_4)_{0.004}$ sample, EDS line scanning is performed for the matrix cross the

nanoprecipitates (Supplementary Fig. 9). The content of S in the nanoprecipitate is less than that in the matrix, whereas the Se content is slightly higher, indicating that the coherent nanoprecipitates are the Cu-Se-S solid solution. Besides, there is only one set of diffraction spots in the region that contains both Cu₂S matrix and Cu-Se-S compound. Previous study has proved that the coherent interfaces can generate between Cu₂S and Cu-Se-S solid solution [Inorg. Chem. 60, 13269-13277 (2021)].

Supplementary Fig. 9 TEM characterization of the coherent interfaces in the $\text{Cu}_{1.992}\text{SSe}_{0.016}(\text{Cu}_2\text{SnSe}_4)_{0.004}$ sample. (a) HRTEM of the Cu-Se-S compound and copper sulfide matrix. (b) EDS element line scanning cross the Cu-Se-S compound as shown in (a). (c) SAED of the area that containing both Cu₂S matrix and Cu-Se-S solid solution.

The interfaces between Cu₂S matrix and Cu₂SnSe₄ precipitate are also observed, the corresponding HRTEM image is shown in Supplementary Fig. 8. There are transition regions of atomic disorder between the Cu₂S matrix and Cu₂SnSe₄ precipitate, indicating that the amorphous interfaces exist between Cu₂S and Cu₂SnSe₄ due to the obviously different crystal structure and lattice parameters. Extra interfaces can block the carrier transport but also effectively scatter the phonon.

The related results have been added in the supplementary information as the Supplementary Fig. 8 and Supplementary Fig. 9, respectively, and the corresponding discussions have been added in the revised manuscript.

Supplementary Fig. 8 TEM characterization of interface between Cu₂SnSe₄ and Cu₂S matrix. (a) HAADF image of Cu_{1.992}SSe_{0.016}(Cu₂SnSe₄)_{0.004} sample, showing the existence of nanoprecipitates. (b) HRTEM of the Cu₂SnSe₄ precipitate and Cu₂S matrix, amorphous interfaces exist between two phases. SAED of (c) Cu₂S matrix and (d) Cu₂SnSe₄ precipitate.

5. While multiple factors are changing simultaneously, including structure (monoclinic/tetragonal), composition (Cu vacancies), precipitates, and nanopores, how did the authors understand which is the key factor influencing the performance? It would be much more trustworthy if the authors could realize better control of these factors. At least, more detailed discussions about the influence of these factors should be provided.

Response: Thanks for your valuable suggestion. Cu vacancies are the key factors of optimizing the electrical conductivity and stability of copper sulfide-based materials, whereas the multiscale lattice defects act as the key roles in overall reducing lattice thermal conductivity of the materials. Compositional regulation and structural evolution are simultaneously realized by the two-step

optimization, benefiting for synergistically improving thermoelectric performance.

For superionic conductors, composition off-stoichiometry adjustment is more effective in tuning carrier concentration than doping due to the low solid solubility limit. Therefore, the carrier concentration of copper sulfides can vary in a few orders of magnitude on increasing the contents of Cu vacancies by adding extra Se.

Various lattice defects contribute to scatter low-to-mid frequency phonon, causing the significantly decreased thermal conductivity. In particular, the dislocation cores and strain can strongly scatter phonon with mid frequencies to reduce the lattice thermal conductivity at medium temperature, endowing copper sulfides higher thermoelectric performance. Additionally, the extra grain boundaries and pore interfaces benefit to strengthen the scattering of phonon with low frequencies for lowering the lattice thermal conductivity at low temperature.

Monoclinic Cu_2S and tetragonal $\text{Cu}_{1.96}\text{S}$ coexist in the $\text{Cu}_{1.992}\text{S}(\text{Cu}_2\text{SnSe}_4)_{0.004}$ and $\text{Cu}_{1.992}\text{SSe}_{0.016}(\text{Cu}_2\text{SnSe}_4)_{0.004}$ samples, different phase structures affect the turning points of the electrical and thermal transport properties. The increased content of tetragonal $\text{Cu}_{1.96}\text{S}$ results in the enlarged temperature window for achieving high power factor, benefiting for improving average thermoelectric performance.

The key factors of affecting the electrical and thermal performance are highlighted in the summary, and the specific discussions have been added in the corresponding parts in the revised manuscript.

6. Limited by technique accuracy, employing room-temperature Hall data for carrier transport analysis is understandable. However, the temperature inconsistency should at least be stated.

Response: Thanks for your valuable suggestion. Copper sulfides are degenerate semiconductors, thus, there is only a slight variation in carrier concentration for $\text{Cu}_{1.992}\text{SSe}_{0.016}(\text{Cu}_2\text{SnSe}_4)_{0.004}$ with

temperature. The related description has been added in the revised manuscript.

Additionally, the n_H and μ_H for the $\text{Cu}_{1.992}\text{SSe}_{0.016}(\text{Cu}_2\text{SnSe}_4)_{0.004}$ sample within a temperature range of 323-873 K were also characterized (Supplementary Fig. 14). Carrier mobility gradually drops with temperature due to the enhanced inhabitation of carriers by the lattice vibration and Cu ion migration. The related results have been added in the revised supplementary information.

Supplementary Fig. 14 Hall measurement for $\text{Cu}_{2-2x}\text{SSe}_{4x}(\text{Cu}_2\text{SnSe}_4)_x$ ($x=0, 0.003, 0.0035, 0.004, 0.005$ and 0.006) bulk specimen at 300 K to 873 K. Temperature dependence of (a) electrical conductivity, (b) carrier concentration, and (c) carrier mobility of all bulk composites.

7. From the performance data, I can hardly see any result supporting the claim of ‘decoupled carrier phonon transport’. Although, a low lattice thermal conductivity is approached, the high electrical

performance is mainly due to high carrier concentration. Instead, with step-by-step optimization, the carrier mobility are continuously decreasing. Please think more carefully about term-use, and avoid misuse, or provide straight-forward evidence/results.

Response: Thanks for your valuable suggestion. We agree with the reviewer's comments. The coupled thermoelectric parameters, such as electrical conductivity and Seebeck coefficient, carrier mobility and lattice thermal conductivity, have not been optimized simultaneously by the step-by-step optimization. Therefore, the related term as "decoupled" has been moved, the related descriptions have been modified in the revised manuscript. Additionally, the title of this manuscript has also been modified as "**Synergistically Optimized** Electron and Phonon Transport in High-performance Copper Sulfides Thermoelectric Materials via One-pot Modulation of the Second Phases and Cu Vacancies".

Reviewer #2 (Remarks to the Author):

This manuscript describes the thermoelectric properties of p-type Cu₂S. The authors claim that a method to simultaneously achieve precipitation and Cu vacancy modulation is proposed, effectively decoupling electron and phonon transport in copper sulfide, and achieving a high ZT of 1.6 at 873 K. However, the effects of precipitates and copper vacancies on the thermoelectric properties of Cu₂S have been published in Nano Energy 49, 267-273 (2018) and J Mater. Sci-Mater. El. 30, 5177-5184 (2019), and the ZT of Cu₂S has exceeded 2.1 by controlling the microstructure (Angew. Chem. Int. Ed., 2022, 61.45: e202212885.). In addition, some of the experimental observations need clear explanations, as shown below:

Response: We thank the referee 2 for his/her over all positive comments and suggestions. The significant advances in this work are that this is the first time to introduce the in-situ generated second phase and Cu vacancies in Cu-S system via one-pot approach, and the power factor, lattice thermal

conductivity and electrical stability were simultaneously optimized, resulting in the record high ZT of 1.6 at 873 K for the $\text{Cu}_{1.96}\text{S}$ -based thermoelectric materials. The difference between this study and the previous works has been compared, which has been added in the introduction part in the revised manuscript. Previous works have investigated that compositing with multi-walled carbon nanotube [*J. Mater. Sci-Mater. El.* 30, 5177-5184 (2019)] or grapheme [*Nano Energy* 49, 267-273 (2018)] are effective in improving thermoelectric performance of copper sulfides by reducing thermal conductivity. Additionally, Cu_{2-x}S phase-junction nanocomposites with superior thermoelectric performance can be synthesized by surface-ligand tuning [*Angew. Chem. Int. Ed.* 61, e2022128852022, (2022)], indicating that carefully adjusting Cu vacancies benefits to optimize the thermoelectric properties. Nevertheless, the electrical stability of these materials has not been reported, and the method of spontaneously introducing Cu vacancies and in-situ precipitates according to the designed composition by one-pot approach have not reported yet.

1. The description of the increase in Cu vacancies caused by excess Se requires more experimental or theoretical research, since Cu vacancies play an important role in the thermoelectric and microstructure of Cu_2S in this work.

Response: Thanks for your valuable suggestion. In order to characterize the effect of additional Se on the compositional evolution of the material, EPMA analyses were performed for the $\text{Cu}_{1.992}\text{S}(\text{Cu}_2\text{SnSe}_4)_{0.004}$ and $\text{Cu}_{1.992}\text{SSe}_{0.016}(\text{Cu}_2\text{SnSe}_4)_{0.004}$ samples, respectively (Supplementary Fig. 5). The ratio of Cu and S/Se in the $\text{Cu}_{1.992}\text{S}(\text{Cu}_2\text{SnSe}_4)_{0.004}$ sample is close to the pristine Cu_2S , indicating that the introduction of the designed content of Sn and Se in the Cu_2S would not obviously change the stoichiometric ratio of main phase. Furthermore, the introduction of extra Se without changing other condition can significantly decrease the Cu content of the matrix, since that the

generation of Cu-Se-S solid solution consumes with Cu of the raw materials. Therefore, the ratio of Cu and S/Se for the $\text{Cu}_{1.992}\text{S}(\text{Cu}_2\text{SnSe}_4)_{0.004}$ sample reduces to about 1.96. Related descriptions have been added in the supplementary information.

Supplementary Fig. 5 EPMA results of the typical samples. Backscattered electron (BSE) images of the (a) $\text{Cu}_{1.992}\text{S}(\text{Cu}_2\text{SnSe}_4)_{0.004}$ and (b) $\text{Cu}_{1.992}\text{SSe}_{0.016}(\text{Cu}_2\text{SnSe}_4)_{0.004}$ sample, (c) the content of various elements in these two specimens by EPMA.

2. The composition of the precipitate needs double check. Only one Bragg peak shown in Fig. 2, it is hard to index Cu_2SnSe_4 . Also, the XRD patterns of Cu_2SnSe_4 and Cu_2SnSe_3 are nearly the same. Thus, more study is needed to verify the precipitates.

Response: Thanks for your suggestion. We agree that it is hard to prove the existence of Cu_2SnSe_4 in the samples only by XRD due to the overlap of the diffraction peaks. Therefore, TEM characterizations of the $\text{Cu}_{1.992}\text{SSe}_{0.016}(\text{Cu}_2\text{SnSe}_4)_{0.004}$ sample were performed. Nanoprecipitates exist in the $\text{Cu}_{1.992}\text{SSe}_{0.016}(\text{Cu}_2\text{SnSe}_4)_{0.004}$ sample, which can be confirmed as the Cu-Sn-Se compound by EDS mapping (Supplementary Fig. 7 (c-f)). Additionally, the HRTEM of the nanoprecipitate and the corresponding SAED are shown in Supplementary Fig. 7 (b). The lattice spacing of 0.21 nm is

consistent with the (220) plane of cubic Cu_2SnSe_4 . Therefore, the introduction of the suitable content of Sn and Se in copper sulfide can produce the Cu_2SnSe_4 nanoprecipitates. The related results have been added as Supplementary Fig. 7 in the revised supporting information and the corresponding discussions have been added in the revised manuscript.

Supplementary Fig. 7 TEM characterization for the typical precipitate. (a) HAADF image of $\text{Cu}_{1.992}\text{SSe}_{0.016}(\text{Cu}_2\text{SnSe}_4)_{0.004}$ sample, showing the existence of nanoprecipitates. (b) HRTEM of the precipitate with the inset of the corresponding SAED. EDS element mapping of the area in (a), (c) is Cu, (d) is Sn, (e) is Se and (f) is S.

3. Fig. 3, the nanoprecipitates shown in C is not the one shown in A, thus a detailed analysis of the precipitate shown in C is needed. The crystal structure of Cu_2S and Cu_2SnSe_4 or Cu_2SnSe_3 is largely different, it is hard to get a coherent boundary between them.

Response: We are sorry for lacking the detailed evidence and explanation in the previous manuscript that makes you confuse. The nanoprecipitates shown in Fig. 3c1 is not Cu_2SnSe_4 or Cu_2SnSe_3 , which

is actually the Cu-Se-S solid solution. The prepared sample for TEM has been oxidized, thus we prepare the new specimen and perform the TEM observation. Related results have been added in the supporting information as Supplementary Fig. 9. The coherent interfaces exist between the copper sulfide matrix and the Cu-Se-S compounds. This type of nanoprecipitates can be observed elsewhere in the $\text{Cu}_{1.992}\text{SSe}_{0.016}(\text{Cu}_2\text{SnSe}_4)_{0.004}$ sample, EDS line scanning is performed for the matrix cross the nanoprecipitates (Supplementary Fig. 9). The content of S in the nanoprecipitate is less than that in the matrix, whereas the Se content is slightly higher, indicating that the coherent nanoprecipitates are the Cu-Se-S solid solution. Besides, there is only one set of diffraction spots in the region that contains both Cu_2S matrix and Cu-Se-S compound. Previous study has proved that the coherent interfaces can generate between Cu_2S and Cu-Se-S solid solution [*Inorg. Chem.* 60, 13269-13277 (2021)].

Supplementary Fig. 9 TEM characterization of the coherent interfaces in the $\text{Cu}_{1.992}\text{SSe}_{0.016}(\text{Cu}_2\text{SnSe}_4)_{0.004}$ sample. (a) HRTEM of the Cu-Se-S compound and copper sulfide matrix. (b) EDS element line scanning cross the Cu-Se-S compound as shown in (a). (c) SAED of the area that containing both Cu_2S matrix and Cu-Se-S solid solution.

The interfaces between Cu_2S matrix and Cu_2SnSe_4 precipitate are also observed, the corresponding HRTEM image is shown in Supplementary Fig. 8. There are transition regions of atomic

disorder between the Cu_2S matrix and Cu_2SnSe_4 precipitate, indicating that the amorphous interfaces exist between Cu_2S and Cu_2SnSe_4 due to the obviously different crystal structure and lattice parameters. Extra interfaces can block the carrier transport but also effectively scatter the phonon. Related results have been added as the Supplementary Fig. 8 in the supporting information, and the corresponding discussions have been added in the revised manuscript.

Supplementary Fig. 8 TEM characterization of interface between Cu_2SnSe_4 and Cu_2S matrix. (a) HAADF image of $\text{Cu}_{1.992}\text{SSe}_{0.016}(\text{Cu}_2\text{SnSe}_4)_{0.004}$ sample, showing the existence of nanoprecipitates. (b) HRTEM of the Cu_2SnSe_4 precipitate and Cu_2S matrix, amorphous interfaces exist between two phases. SAED of (c) Cu_2S matrix and (d) Cu_2SnSe_4 precipitate.

The Cu_2SnSe_4 precipitates and Cu-Se-S solid solution were clearly distinguished in the Fig. 3 in the revised manuscript.

Fig. 3 STEM characterization of the $\text{Cu}_{1.992}\text{SSe}_{0.016}(\text{Cu}_2\text{SnSe}_4)_{0.004}$ bulk specimen. (a) HAADF image exhibits the existence of precipitates and nanopores in the material. **(b)** EDS element mapping of the area in (a), indicating that one of the precipitates in the bulk composite is Cu_2SnSe_4 . **(c)** High-resolution TEM (HRTEM) image of the $\text{Cu}_{1.992}\text{SSe}_{0.016}(\text{Cu}_2\text{SnSe}_4)_{0.004}$ bulk specimen, expressing the existence of a high-density dislocation area and regular precipitates. **(c1)** and **(c2)** are the inverse fast Fourier transform (IFFT) images at the selected area marked by pink and blue squares in (c),

respectively. (c3) schematic diagram of the formation of edge dislocation by Cu vacancy. (d) The stress distribution of the whole region in (c) by geometric phase analysis (GPA), and the color bar represents -10% to 10% strain.

4. Fig. 4, the electrical conductivity and carrier concentration of $\text{Cu}_2\text{S}(\text{SnSe})\text{Se}$ is much higher than that of $\text{Cu}_2\text{S}(\text{SnSe})$, while the Seebeck coefficient of these two samples are nearly the same. This is strange in thermoelectric materials; the authors need to explain it clearly.

Response: Thanks for your suggestion. The electrical conductivity of $\text{Cu}_{1.992}\text{SSe}_{0.016}(\text{Cu}_2\text{SnSe}_4)_{0.004}$ is much higher than that of $\text{Cu}_{1.992}\text{S}(\text{Cu}_2\text{SnSe}_4)_{0.004}$ owing to the increased carrier concentration caused by the generated Cu vacancies, resulting in the reduction of Seebeck coefficient. In order to clearly figure out the effect of additional Se on adjusting electrical transport properties, the electrical conductivity and Seebeck coefficient of the $\text{Cu}_{1.992}\text{S}(\text{Cu}_2\text{SnSe}_4)_{0.004}$ and $\text{Cu}_{1.992}\text{SSe}_{0.016}(\text{Cu}_2\text{SnSe}_4)_{0.004}$ specimens are retested, related results are shown in the Fig.4 in the revised manuscript. The σ of $\text{Cu}_{1.992}\text{SSe}_{0.016}(\text{Cu}_2\text{SnSe}_4)_{0.004}$ is further improved compared to that of the $\text{Cu}_{1.992}\text{S}(\text{Cu}_2\text{SnSe}_4)_{0.004}$ sample, since that the hole concentration of the material increases by introducing more Cu vacancies. The n_H reaches $11.09 \times 10^{20} \text{ cm}^{-3}$ for the $\text{Cu}_{1.992}\text{SSe}_{0.016}(\text{Cu}_2\text{SnSe}_4)_{0.004}$ sample, which is dramatically higher than that of both the pristine Cu_2S sample ($0.08 \times 10^{20} \text{ cm}^{-3}$) and $\text{Cu}_{1.992}\text{S}(\text{Cu}_2\text{SnSe}_4)_{0.004}$ sample ($2.57 \times 10^{20} \text{ cm}^{-3}$). Additionally, the Seebeck coefficient (S) of $\text{Cu}_{1.992}\text{S}(\text{Cu}_2\text{SnSe}_4)_{0.004}$ sample obviously drops by comparison with that of the pure Cu_2S specimen because of the variation in carrier concentration (Fig. 4b). The S further reduces after introducing extra Se, which still remains at a high-level owing to the enhanced carrier effective mass after alloying with Se (Fig. 4e). Previous studies have proved that alloying with Se or Te would slightly increase the carrier effective mass of copper sulfides. Related description has been added in the revised manuscript.

Fig. 4 Electrical transport properties of Cu_2S , $\text{Cu}_{1.992}\text{S}(\text{Cu}_2\text{SnSe}_4)_{0.004}$ and $\text{Cu}_{1.992}\text{SSe}_{0.016}(\text{Cu}_2\text{SnSe}_4)_{0.004}$ bulk samples. Temperature dependence of (a) electrical conductivity, (b) Seebeck coefficient and (f) power factor of Cu_2S , $\text{Cu}_{1.992}\text{S}(\text{Cu}_2\text{SnSe}_4)_{0.004}$ and $\text{Cu}_{1.992}\text{SSe}_{0.016}(\text{Cu}_2\text{SnSe}_4)_{0.004}$ bulk samples. (c) Composition-dependent hall carrier concentration and mobility at 300 K for all bulk samples. (d) $(ahv)^2$ vs. hv of all specimens, the optical band gap (E_g) can be estimated by extrapolating the straight line to $(ahv)^2=0$. (e) Seebeck coefficient as a function of carrier concentration at 300 K and 873 K. Red lines are obtained by the SPB model and estimating the carrier effective mass, and dots are obtained by experimental data.

5. The authors should provide thermoelectric data for at least 3 heating-cooling cycles to point out the stability of this material.

Response: Thanks for your suggestion. The cycling test of electrical conductivity, Seebeck coefficient, power factor and thermal conductivity of the $\text{Cu}_{1.992}\text{SSe}_{0.016}(\text{Cu}_2\text{SnSe}_4)_{0.004}$ sample are performed, the results have been added in the supplementary information as Supplementary Fig. 20. Owing to the introduction of Cu vacancies and precipitates in the copper sulfides, the thermoelectric performance

of the Sn and Se-added samples is maintained in the cycling test. The related descriptions have been added in the revised manuscript.

Supplementary Fig. 20 Cycling measurement of $\text{Cu}_{1.992}\text{SSe}_{0.016}(\text{Cu}_2\text{SnSe}_4)_{0.004}$ bulk specimens.

The temperature dependence of (a) electrical conductivity, (b) Seebeck coefficient and (c) power factor and (d) thermal conductivity of the $\text{Cu}_{1.992}\text{SSe}_{0.016}(\text{Cu}_2\text{SnSe}_4)_{0.004}$ bulk specimens in the cycling measurement.

Reviewer #3 (Remarks to the Author):

In this article, the authors fabricated a series of copper sulfide materials with Cu vacancies and various defects by incorporating Cu_2S with Sn and Se. The high-density dislocations, multiscaled precipitates and pores were simultaneously obtained in the bulk material and the effects of various mechanisms on scattering mid-to-low frequency phonon were discussed. They found the carrier concentration and electrical properties of the copper sulfides were effectively adjusted by spontaneously tuned Cu content.

This work provides an interesting method to improve the thermoelectric performance and stability of copper sulfides. My comments/suggestions about this work are listed below.

Response: We thank the referee 3 for his/her over all positive comments and suggestions. And we have tried our best to revise the manuscript.

(1) The stress distribution in Fig. 3d is interesting. Have the authors performed the geometric phase analysis on the pristine Cu_2S ?

Response: Thanks for your question. The stress distribution of the pristine Cu_2S material is performed by the geometric phase analysis, which has been added in the supporting information as Supplementary Fig. 11. There is slight stress in the pure Cu_2S bulk specimen, which should be ascribed to the tiny content of Cu vacancies in the material. Cu vacancies might exist after SPS process, the deficiency of Cu atoms would result in the extra repulsion force in the lattice. Nevertheless, the stress in the pristine Cu_2S material is weak by comparison of the $\text{Cu}_{1.992}\text{SSe}_{0.016}(\text{Cu}_2\text{SnSe}_4)_{0.004}$ bulk specimen. The related description has been added in the supplementary information.

Supplementary Fig. 11 GPA of the pristine Cu₂S material. (a) HRTEM of the pristine Cu₂S material, the inset is the corresponding SAED, (b-c) stress distribution of the pure Cu₂S material by geometric phase analysis.

(2) In Fig. 4a, the turning points of electrical conductivity gradually shift to the lower temperature with increasing the Sn and Se content. The authors should clarify the reason.

Response: Thanks for your suggestion. The turning points of the electrical conductivity for the Cu_{2-x}SSe_{4x}(Cu₂SnSe₄)_x specimens gradually shift to the lower temperature with addition content, which is mainly attributed to the reduced phase transition temperature of copper sulfides. Tiny content of Cu vacancies are introduced in the Cu_{1.992}S(Cu₂SnSe₄)_{0.004} specimen, the decreased Cu content in the copper sulfides causes the reduced temperature of phase transition. The Cu content of the Cu_{1.992}SSe_{0.016}(Cu₂SnSe₄)_{0.004} sample is further decreased; thus, the lower phase transition temperature promotes the turning points of the electrical conductivity shifting to the lower temperature.

(3) The mechanism for the formation of nanopores is a little obscure? More discussions are required.

Response: Thanks for your suggestion. Nanopores are introduced in the copper sulfides after adding Sn and Se, which is mainly caused by the sulfur volatilization of the materials during the SPS process. Previous studies have proved that the pores within the grains are usually introduced by facilitating the S volatilization, additional nanoprecipitates prefer to distribute into the nanopores. Herein, the structure evolution of the copper sulfides results from the compositional regulation. Cu_{1.96}S, Cu₂SnSe₄ precipitates and Cu-Se-S solid solutions coexist in the material, nanopores generate due to the S volatilization of the Cu_{1.96}S. There are tiny content of larger pores along the grain boundaries, which might be ascribed to the different thermal expansion between Cu_{1.96}S matrix and Cu₂SnSe₄ precipitates. Related descriptions have been added in the revised supplementary information.

(4) What about the thermal stability of the Sn and Se-added samples? The cycling test should be performed.

Response: Thanks for your question. The cycling test of electrical conductivity, Seebeck coefficient, power factor and thermal conductivity of the $\text{Cu}_{1.992}\text{SSe}_{0.016}(\text{Cu}_2\text{SnSe}_4)_{0.004}$ sample are performed, the results have been added in the supplementary information. Owing to the introduction of Cu vacancies and precipitates in the copper sulfides, the thermoelectric performance of the Sn and Se-added samples is maintained in the cycling test, suggesting the excellent thermal stability of the samples. The related descriptions have been added in the revised manuscript.

Supplementary Fig. 20 Cycling measurement of $\text{Cu}_{1.992}\text{SSe}_{0.016}(\text{Cu}_2\text{SnSe}_4)_{0.004}$ bulk specimens.

The temperature dependence of (a) electrical conductivity, (b) Seebeck coefficient and (c) power factor and (d) thermal conductivity of the $\text{Cu}_{1.992}\text{SSe}_{0.016}(\text{Cu}_2\text{SnSe}_4)_{0.004}$ bulk specimens in the cycling measurement.

(5) In Figure 5, the shear sound velocity gradually decreases with Sn and Se content, the authors should explain the reason.

Response: Thanks for your suggestion. The decreased shear sound velocity of the $\text{Cu}_2\text{SSe}_{4x}(\text{Cu}_2\text{SnSe}_4)_x$ specimens with increased Sn and Se content results from the weakened bonding. Extra Se assists to form the solid solution with Cu and S, Se-alloying has been proved to be effective in reducing bonding energy of the Cu_2S materials, resulting in the decreased speed of sound and therefore the lower lattice thermal conductivity. Related description has been added in the revised manuscript.

(6) In Figure 3c, two special nanostructures are marked by “g1” and “g2”, respectively. However, there are no corresponding figure numbers.

Response: Thanks for your comments. The Fig. 3 is renumbered, which has been modified in the revised manuscript.

Fig. 3 STEM characterization of the $\text{Cu}_{1.992}\text{SSe}_{0.016}(\text{Cu}_2\text{SnSe}_4)_{0.004}$ bulk specimen. (a) HAADF image exhibits the existence of precipitates and nanopores in the material. (b) EDS element mapping of the area in (a), indicating that one of the precipitates in the bulk composite is Cu_2SnSe_4 . (c) High-resolution TEM (HRTEM) image of the $\text{Cu}_{1.992}\text{SSe}_{0.016}(\text{Cu}_2\text{SnSe}_4)_{0.004}$ bulk specimen, expressing the existence of a high-density dislocation area and regular precipitates. (c1) and (c2) are the inverse fast Fourier transform (IFFT) images at the selected area marked by pink and blue squares in (c),

respectively. (c3) schematic diagram of the formation of edge dislocation by Cu vacancy. (d) The stress distribution of the whole region in (c) by geometric phase analysis (GPA), and the color bar represents -10% to 10% strain.

(7) Have the authors measured the DSC of the Sn and Se-added samples? Will adding Sn and Se change the phase transition behavior?

Response: Thanks for your question. The DSC curves of the pristine Cu_2S and $\text{Cu}_{1.992}\text{SSe}_{0.016}(\text{Cu}_2\text{SnSe}_4)_{0.004}$ samples in the temperature range of 310-773 K are obtained by the Netzsch STA 449 F3 (Germany) instrument under flowing Ar gas, and the heating speed is 10 K min^{-1} . Related descriptions have been added in the Experimental section of the revised manuscript.

There are two exothermic peaks at about 377 K and 720 K for the pure Cu_2S sample, corresponding to the monoclinic-hexagonal transition (Cu_2S M to H) and the hexagonal-cubic transition (Cu_2S H to C), respectively. Extra Sn and Se shifts these two phase transitions to the lower temperature due to the introduction of Cu vacancies. Additionally, an extra exothermic peak appears at about 418 K, which is consistent with the tetragonal-hexagonal transition. Although alloying with large content of Se in copper sulfides can remove the phase transition from the trigonal phase (T) to the hexagonal phase [*J. Mater. Chem. A* 5, 18148 (2017)], the adding content of Se in this study is tiny, and this phenomenon is not obvious. Therefore, adding Sn and Se can shift the two phase transitions (Cu_2S M to H and Cu_2S H to C) of copper sulfides to the lower temperature and introduce an extra phase transition ($\text{Cu}_{1.96}\text{S}$ T to H). The related results have been added in the supplementary information as Supplementary Fig. 12.

Supplementary Fig. 12 Differential scanning calorimetry (DSC) analysis of Cu_2S and $\text{Cu}_{1.992}\text{SSe}_{0.016}(\text{Cu}_2\text{SnSe}_4)_{0.004}$ samples in the temperature range of 310-773 K.

(8) What about the uncertainty of the experimental data in this study? The error bars should be added in the ZT results.

Response: Thanks for your suggestions. The absolute error of all experimental data can be assumed to be 3%~4%, so the error bar of ZT would be 15%~20%. The error bars of the ZT results have been added in Fig.6a in the revised manuscript.

Fig. 6 Thermoelectric performance of copper sulfide-based bulk composites. Temperature dependence of (a) dimensionless figure of merit ZT of Cu_2S , $\text{Cu}_{1.992}\text{S}(\text{Cu}_2\text{SnSe}_4)_{0.004}$ and $\text{Cu}_{1.992}\text{SSe}_{0.016}(\text{Cu}_2\text{SnSe}_4)_{0.004}$ bulk samples. (b) Carrier concentration dependence of the ZT value at different temperatures by using the SPB model. (c) Experimental and estimated conversion efficiency as a function of temperature gradient for the $\text{Cu}_{1.992}\text{SSe}_{0.016}(\text{Cu}_2\text{SnSe}_4)_{0.004}$ sample. (d) Related electrical resistivity (R/R_0) of the $\text{Cu}_{1.992}\text{SSe}_{0.016}(\text{Cu}_2\text{SnSe}_4)_{0.004}$, $\text{Cu}_{1.96}\text{S}$ and Cu_2S samples with increased current density at 873 K. (e) The critical voltage (V_c) of the pristine Cu_2S and that with Sn and Se addition at 873 K.

(9) In Fig. 6d, the authors evaluated the stability of their samples by measuring the relative resistance variation under different current density. Actually, the critical voltage is a more intrinsic parameter to evaluate the stability of superionic conductors. More details can be found in NATURE COMMUNICATIONS (2018) 9:2910.

Response: Thanks for your suggestions. We agree with the reviewer's suggestion that the critical

voltage is indeed more reasonable to evaluate the electrical stability of the superionic conductors. According to the suggested literature, the critical voltage (V_c) of the $\text{Cu}_{1.992}\text{SSe}_{0.016}(\text{Cu}_2\text{SnSe}_4)_{0.004}$ and $\text{Cu}_{1.992}\text{S}(\text{Cu}_2\text{SnSe}_4)_{0.004}$ is calculated, the results have been added as the inset in the Fig. 9d in the revised manuscript. Furthermore, in order to more reasonably evaluate the electrical stability of the superionic conductors, the critical voltage (V_c) of the $\text{Cu}_{1.992}\text{SSe}_{0.016}(\text{Cu}_2\text{SnSe}_4)_{0.004}$ and $\text{Cu}_{1.992}\text{S}(\text{Cu}_2\text{SnSe}_4)_{0.004}$ samples at 873 K is shown in Fig. 6e, the calculated details can be seen in the reported works [Nat. Commun. **9**, 2910 (2018)]. V_c of pristine Cu_2S is slightly lower than the reported result due to the higher measurement temperature. Besides, higher V_c of $\text{Cu}_{1.992}\text{SSe}_{0.016}(\text{Cu}_2\text{SnSe}_4)_{0.004}$ indicates that the spontaneously generated Cu vacancies and multiscale precipitates contribute to improve the electrical stability of copper sulfides. Related description has been added in the revised manuscript.

Fig. 6 Thermoelectric performance of copper sulfide-based bulk composites. Temperature dependence of (a) dimensionless figure of merit ZT of Cu_2S , $\text{Cu}_{1.992}\text{S}(\text{Cu}_2\text{SnSe}_4)_{0.004}$ and

$\text{Cu}_{1.992}\text{SSe}_{0.016}(\text{Cu}_2\text{SnSe}_4)_{0.004}$ bulk samples. (b) Carrier concentration dependence of the ZT value at different temperatures by using the SPB model. (c) Experimental and estimated conversion efficiency as a function of temperature gradient for the $\text{Cu}_{1.992}\text{SSe}_{0.016}(\text{Cu}_2\text{SnSe}_4)_{0.004}$ sample. (d) Related electrical resistivity (R/R_0) of the $\text{Cu}_{1.992}\text{SSe}_{0.016}(\text{Cu}_2\text{SnSe}_4)_{0.004}$, $\text{Cu}_{1.96}\text{S}$ and Cu_2S samples with increased current density at 873 K. (e) The critical voltage (V_c) of the pristine Cu_2S and that with Sn and Se addition at 873 K.

List of changes

1. The title of this manuscript has been slightly modified according to the requirement by reviewer.
2. More literatures have been cited in the introduction part.
3. EPMA results have been added in Supplementary Fig. 12.
4. More TEM characterizations and discussions have been added.
5. The error bars of ZT values have been added in Fig. 6a.
6. The critical voltage of the samples was added in the Fig. 6d.
7. The DSC results were added in Supplementary Fig. 12.
8. The cycling measurement results of the optimum specimen were added in Supplementary Fig. 20.

REVIEWER COMMENTS

Reviewer #1 (Remarks to the Author):

I appreciate that the authors have tried their best to solve the reviewer comments. However, my key concern remains unsolved. My key concern is still the claimed coherent boundary:

As stated in my previous comments: 'This is highly possibly just a misunderstanding, where there is another thick layer of Cu₂S covering the Cu₂SnSe₄ precipitates, presenting the Cu₂S lattice.'

I agree there are Cu₂SnSe₄ precipitates, as consistent with the EDS results. However, the precipitates are highly possibly just covered by the matrix without really exposed, and it is too thin comparing with the matrix in the characterization region. My viewpoint is consistent with EDS, which the authors used to convince me is insufficient. Another information the authors used to convince me is claimed the SAED, however, to me, considering the intense noise, it looks more like FFT instead. FFT is a mathematic process of the HRTEM image which will definitely be consistent with the HRTEM. However, the problem here is the sample preparation, not data process. Even it is SAED, if the sample is not well-processed, the true information still can be hidden. These information are not trustable enough to convince me. For this reason, I insist this kind claim is misleading and detrimental to the overall thermoelectric field. Scientific claims in Nat. Comm. should be careful and well-evidenced. I would not recommend publication of this study in Nat. Comm.

One additional minor concern:

I agree with the authors that Se doping/alloying can increase Cu vacancies. However, if Cu vacancy is the target, it is confusing why the authors do not try to realize this goal by directly reducing the content of Cu?

Reviewer #2 (Remarks to the Author):

The authors has revised all the raised comments in the resubmitted manuscript, which is acceptable now.

Reviewer #3 (Remarks to the Author):

The authors have satisfactorily answered all the comments in my last round review. I recommend it for publication without further review.

Reviewer #3's further Comments on the report of Reviewer #1

I have carefully read the reports of the Reviewer #1 in the first and second rounds of review. I found the Reviewer #1 thought this work is comprehensive and worth for publication in general. This is consistent with my viewpoint. Now his main concern is whether the Cu₂SnSe₄ and Cu₂S are coherent or not. I agree with him that this issue has not been clearly characterized right now. I suggest to ask the authors to re-characterize the Cu₂SnSe₄/Cu₂S boundaries and discuss this issue more clearly. If Cu₂SnSe₄ and Cu₂S are not coherent, the authors can delete the related contents and discussions in the resubmitted manuscript.

REVIEWER COMMENTS

Reviewer #1 (Remarks to the Author):

I appreciate that the authors have tried their best to solve the reviewer comments. However, my key concern remains unsolved. My key concern is still the claimed coherent boundary:

As stated in my previous comments: 'This is highly possibly just a misunderstanding, where there is another thick layer of Cu_2S covering the Cu_2SnSe_4 precipitates, presenting the Cu_2S lattice.'

I agree there are Cu_2SnSe_4 precipitates, as consistent with the EDS results. However, the precipitates are highly possibly just covered by the matrix without really exposed, and it is too thin comparing with the matrix in the characterization region. My viewpoint is consistent with EDS, which the authors used to convince me is insufficient. Another information the authors used to convince me is claimed the SAED, however, to me, considering the intense noise, it looks more like FFT instead. FFT is a mathematic process of the HRTEM image which will definitely be consistent with the HRTEM. However, the problem here is the sample preparation, not data process. Even it is SAED, if the sample is not well-processed, the true information still can be hidden. These information are not trustable enough to convince me.

For this reason, I insist this kind claim is misleading and detrimental to the overall thermoelectric field. Scientific claims in Nat. Comm. should be careful and well-evidenced. I would not recommend publication of this study in Nat. Comm.

Response: We thank the referee 1 for his/her valuable comments and suggestions. And we have tried our best to revise the manuscript. We have re-analyzed all the TEM results with the kind help of Dr. Lei Jin at Forschungszentrum Jülich, and agree with the reviewer that the interfaces between the Cu_2SnSe_4 precipitates and Cu_2S matrix are not coherent. We also agree that the precipitates are highly

possibly just covered by the matrix without really exposed, and it is too thin comparing with the matrix in the characterization region. The related discussion about the coherent interfaces has been removed. The precipitates in the matrix could scatter the phonon to reduce lattice thermal conductivity whatever the interfaces between the precipitates and matrix are coherent or not. The TEM images in main text and also supplementary information are all reorganized and discussed.

Fig. 3 STEM characterization of the $\text{Cu}_{1.992}\text{SSe}_{0.016}(\text{Cu}_2\text{SnSe}_4)_{0.004}$ bulk specimen. (a) HAADF image exhibits the existence of precipitates and nanopores in the material. (b) EDS element mapping of the area in (a), indicating that precipitates in the bulk composite are Cu_2SnSe_4 . (c) High-resolution TEM (HRTEM) image of the $\text{Cu}_{1.992}\text{SSe}_{0.016}(\text{Cu}_2\text{SnSe}_4)_{0.004}$ bulk specimen, expressing the existence

of a high-density dislocation area. (d) Corresponding fast Fourier transform (FFT) image of the area, (e) the inverse fast Fourier transform (IFFT) image at the selected area. (f) The stress distribution of the whole region in (c) by geometric phase analysis (GPA), and the color bar represents -10% to 10% strain. (g) Schematic diagram of the formation of edge dislocation by Cu vacancy.

We are sorry to claim the fast Fourier transform (FFT) image as the selection area electron diffraction (SAED) pattern, the related term-use has been carefully checked in the revised manuscript. TEM was performed for the $\text{Cu}_{1.992}\text{SSe}_{0.016}(\text{Cu}_2\text{SnSe}_4)_{0.004}$ sample, nanoprecipitates were observed in the TEM image, which consist of Cu, Sn and Se by EDS mapping. Therefore, the introduction of the suitable content of Sn and Se in copper sulfide can produce the Cu_2SnSe_4 nanoprecipitates. But the precipitates are highly possibly just covered by the matrix without really exposed, and it is too thin comparing with the matrix in the characterization region. It is therefore hard to clearly show the lattice of the Cu_2SnSe_4 from FFT pattern and/or HRTEM fringe due to the strong effects of the matrix lattices. It can be reasonable proposed that the interfaces between Cu_2S and Cu_2SnSe_4 are incoherent due to their different crystal structures. The related results have been modified in the supplementary information as Supplementary Fig. 5, and the corresponding discussions have been added in the revised manuscript.

Supplementary Fig. 5 TEM characterization for the typical precipitate. (a) HAADF image of $\text{Cu}_{1.992}\text{SSe}_{0.016}(\text{Cu}_2\text{SnSe}_4)_{0.004}$ sample, showing the existence of nanoprecipitates. EDS mapping of the area in (a), (b) is Cu, (c) is Sn, (d) is Se and (e) is S.

Actually, the copper sulfides sample preparation for TEM is still a technological problem, it is hard to prepare a very thin specimen for copper sulfide-based materials by the ion thinning due to the brittleness. Additionally, using focused ion beam technique (FIB) to prepare the sample would change the composition and microstructure of copper sulfides, and the sample are usually not resistant to irradiation.

One additional minor concern:

I agree with the authors that Se doping/alloying can increase Cu vacancies. However, if Cu vacancy is the target, it is confusing why the authors do not try to realize this goal by directly reducing the content of Cu?

Response: Thanks for your questions. Increasing Cu vacancies is our target but not the only one, introducing excessive Se can not only increase the Cu vacancies, but also to decrease the bonding energy by Se alloying and to introduce the precipitates. The reduced bonding energy of Cu_2S can

weaken the sound velocity of materials, and the introduced precipitates are beneficial for further suppressing lattice thermal conductivity by enhancing phonon scattering, consequently leading the improved thermoelectric properties.

Reviewer #2 (Remarks to the Author):

The authors has revised all the raised comments in the resubmitted manuscript, which is acceptable now.

Response: Thank you for your recognition. We are glad to address your concerns for this work.

Reviewer #3 (Remarks to the Author):

The authors have satisfactorily answered all the comments in my last round review. I recommend it for publication without further review.

Response: Thank you for your recognition. We are glad to address your concerns for this work.

Reviewer #3's further Comments on the report of Reviewer #1

I have carefully read the reports of the Reviewer #1 in the first and second rounds of review. I found the Reviewer #1 thought this work is comprehensive and worth for publication in general. This is consistent with my viewpoint. Now his main concern is whether the Cu_2SnSe_4 and Cu_2S are coherent or not. I agree with him that this issue has not been clearly characterized right now. I suggest to ask the authors to re-characterize the $\text{Cu}_2\text{SnSe}_4/\text{Cu}_2\text{S}$ boundaries and discuss this issue more clearly. If Cu_2SnSe_4 and Cu_2S are not coherent, the authors can delete the related contents and discussions in the resubmitted manuscript.

Response: We thank the referee 3 for his/her valuable comments and suggestions. We have re-analyzed all the TEM results with the kind help of Dr. Lei Jin at Forschungszentrum Jülich, and agree with the reviewer that the interfaces between the Cu_2SnSe_4 precipitates and Cu_2S matrix are not coherent. We also agree that the precipitates are highly possibly just covered by the matrix without really exposed, and it is too thin comparing with the matrix in the characterization region. The related discussion about the coherent interfaces has been removed. The precipitates in the matrix could scatter the phonon to reduce lattice thermal conductivity whatever the interfaces between the precipitates and matrix are coherent or not. The TEM images in main text and also supplementary information are all reorganized and discussed.

Fig. 3 STEM characterization of the $\text{Cu}_{1.992}\text{SSe}_{0.016}(\text{Cu}_2\text{SnSe}_4)_{0.004}$ bulk specimen. (a) HAADF image exhibits the existence of precipitates and nanopores in the material. (b) EDS element mapping of the area in (a), indicating that precipitates in the bulk composite are Cu_2SnSe_4 . (c) High-resolution TEM (HRTEM) image of the $\text{Cu}_{1.992}\text{SSe}_{0.016}(\text{Cu}_2\text{SnSe}_4)_{0.004}$ bulk specimen, expressing the existence of a high-density dislocation area. (d) Corresponding fast Fourier transform (FFT) image of the area, (e) the inverse fast Fourier transform (IFFT) image at the selected area. (f) The stress distribution of the whole region in (c) by geometric phase analysis (GPA), and the color bar represents -10% to 10% strain. (g) Schematic diagram of the formation of edge dislocation by Cu vacancy.

List of changes

1. The contents and discussions about the 'coherent interface' have been removed.
2. TEM images have been reorganized in the revised manuscript and supplementary information.
3. Some term-use have been re-checked and modified.

REVIEWERS' COMMENTS

Reviewer #3 (Remarks to the Author):

The authors have satisfactorily answered the comments in my last round review. I recommend it for publication without further review.

REVIEWER COMMENTS

Reviewer #3 (Remarks to the Author):

The authors have satisfactorily answered the comments in my last round review. I recommend it for publication without further review.

Response: We thank the referee 3 for his/her valuable comments. We are glad to address your concerns for this work.